# Open Compound Domain Adaptation with Object Style Compensation for Semantic Segmentation

**Tingliang Feng**[1,2*]    **Hao Shi**[1,3*]    **Xueyang Liu**[1]    **Wei Feng**[1,2]

**Liang Wan**[1]    **Yanlin Zhou**[4]    **Di Lin**[1†]

[1]College of Intelligence and Computing, Tianjin University

[2]Laboratory of Computation and Analytics of Complex Management Systems (CACMS), Tianjin University

[3]Department of Automation, Tsinghua University    [4]Dunhuang Academy

{fengtl, xyliu850569498, lwan}@tju.edu.cn    shi-h23@mails.tsinghua.edu.cn

wfeng@ieee.org    zhouyanlin@dha.ac.cn    Ande.lin1988@gmain.com

## Abstract

Many methods of semantic image segmentation have borrowed the success of open compound domain adaptation. They minimize the style gap between the images of source and target domains, more easily predicting the accurate pseudo annotations for target domain's images that train segmentation network. The existing methods globally adapt the scene style of the images, whereas the object styles of different categories or instances are adapted improperly. This paper proposes the *Object Style Compensation*, where we construct the *Object-Level Discrepancy Memory* with multiple sets of discrepancy features. The discrepancy features in a set capture the style changes of the same category's object instances adapted from target to source domains. We learn the discrepancy features from the images of source and target domains, storing the discrepancy features in memory. With this memory, we select appropriate discrepancy features for compensating the style information of the object instances of various categories, adapting the object styles to a unified style of source domain. Our method enables a more accurate computation of the pseudo annotations for target domain's images, thus yielding state-of-the-art results on different datasets.

## 1   Introduction

The task of open compound domain adaptation (OCDA) for semantic segmentation aims to train the segmentation models [1, 2, 3, 4, 5, 6] using a source domain with pixel-level annotations and target domains of mixed styles with no annotations. The model is trained to perform well on images from the target domains that share the same style as seen during training, as well as on open domain images that are not encountered during training. The recent methods [7, 8, 9, 10, 11] of semantic segmentation utilize the open compound domain adaptation (OCDA), which harnesses the annotated images and the annotation-free images captured in the open environments to train the segmentation network. In this manner, the segmentation network learns from richer data with diverse object appearances while requiring reasonable effort for image labeling. Here, the annotated images belong to the source domain, while the annotation-free images are in the compound target domain for capturing the complexity of open environments.

There is a gap between the image styles of the source and target domains, where the image styles usually are regarded as an array of scene-level properties (e.g., weather and lighting conditions). The

---

[*]Co-first authors. Equal contribution.

[†]Di Lin is the corresponding author of this paper.

37th Conference on Neural Information Processing Systems (NeurIPS 2023).

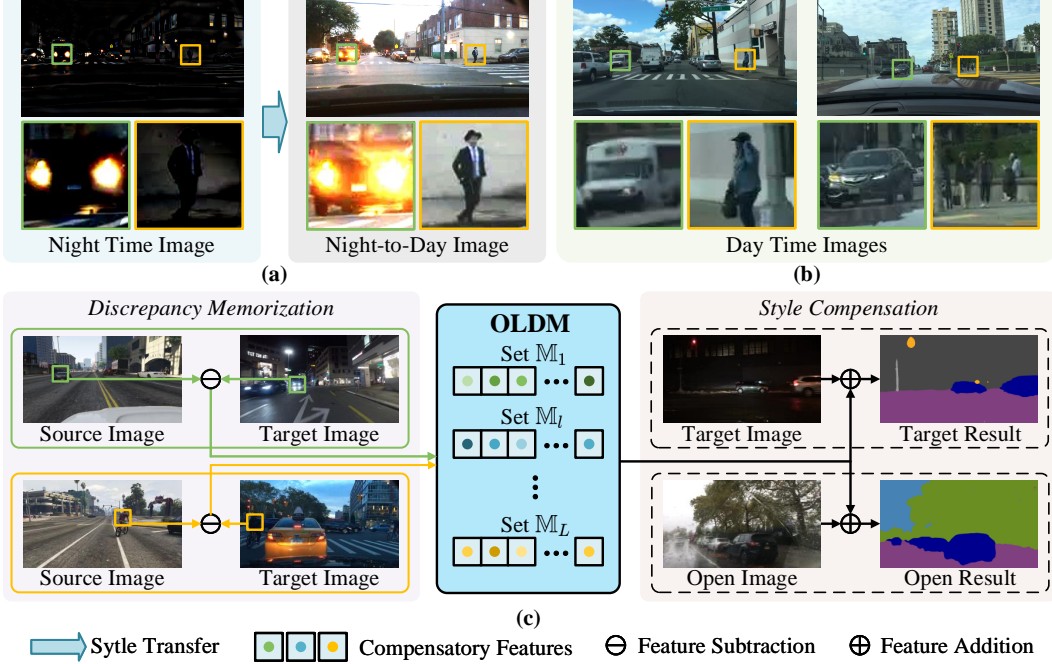

Figure 1: (a) Cars and pedestrians adapted from the night to day scenes. (b) The usual cars and pedestrians in the real day scene. (c) OLDM contains the discrepancy features for representing the difference of object styles in the source and target domains, which are used for adapting the object-level styles from target to source domains.

scene styles are adapted[3] to narrow the gap between image styles, allowing the segmentation network to focus on the intrinsic object appearances of every domain. It helps the network to predict accurate segmentation masks for many images in the compound target domain. The predicted masks play as the pseudo-pixel-wise annotations for tuning the segmentation network.

Typically, the adaptation of scene styles follows the homologous patterns to change the object styles in the same image. It lets various object categories undergo the style change along the same direction (see Figure 1(a), the clear pedestrians and overexposed cars adapted from the night to day scenes). In the common context, the objects in various categories, even different object instances in the same category, may have specific styles within a scene (see Figure 1(b), the usual cars and pedestrians in the day time). Adapting scene styles may yield unreasonable object styles inconsistent with the usual context, leading to problematic annotations for the network training. Additional, The typical OCDA methods learn deep network parameters. The parameters are learned from the source and target images in the training set. But they cannot trivially transform the open domain's object features that contain many unseen styles. Using the learned parameters directly may transform the object features of the open domain to the latent sub-space misaligned with the source domain.

This paper advocates equipping OCDA with object style compensation for semantic segmentation. In contrast to the adaptation of scene styles, the compensation respects the object style discrepancies between the source and target domains. These discrepancies are captured for the independent object categories and instances. Intuitively, the style discrepancies can be memorized during network learning. It enables the appropriate style discrepancies, which can be regarded as the prior information, to be selected and added to the object features. Meanwhile, rather than using the network parameters to transform the target/open domain's features to the source domain, we learn the discrepancy features representing the style differences across domains. Consequently, we compensate the object features to adapt the object styles to the similar style of source domain. The compensation yields consistent object context within the scene, helping the segmentation network to compute reliable pseudo annotations.

---

[3]Someone performs the adaptation in the image or feature space. The latter one is considered in this paper.

Specifically, we conduct the object style compensation for adapting the images from target to source domains. The pipeline comprises *Discrepancy Memorization* and *Style Compensation*, as illustrated in Figure 1(c). During *Discrepancy Memorization*, we construct a feature base, *Object-Level Discrepancy Memory* (OLDM), which consists of multiple feature sets. Each set contains the discrepancy features for the identical object category. Here, we compute the discrepancy feature by subtracting the target domain's object features from the same category's object features of the source domain, letting the discrepancy features represent the difference of the object styles across two domains. We compute the discrepancy features for the object instances in different images of target domain. During *Style Compensation*, given a query object in the target domain's image, we select the discrepancy feature from the feature set of the requested category, where the discrepancy feature represents the instance much relevant to the query object. In this way, the discrepancy feature customizes the information, which is category- and instance-orientated, for compensating the query object's feature. The compensated object features are used for regressing the pseudo annotations.

We conduct an intensive evaluation of the object style compensation. On the public datasets (e.g., C-Driving [7], ACDC [12], Cityscapes [13], KITTI [14], and WildDash [15]) that allows OCDA to assist semantic segmentation, the object style compensation surpasses state-of-the-art methods, demonstrating its effectiveness.

## 2  Related Work

### 2.1  Domain Adaptation for Semantic Segmentation

Numerous studies have been conducted in the realm of unsupervised domain adaptation (UDA), multi-target domain adaptation (MTDA), and domain generalization (DG) for the task of semantic segmentation. These investigations encompass a plethora of approaches and methodologies, aimed at enhancing the performance and robustness of segmentation models. In the domain of UDA, several notable works have contributed to this field [16, 17, 18, 19, 20, 21, 22, 23, 24, 25, 26, 27]. These UDA methods encompass various strategies such as style regression [28], domain adversarial training [18, 29], and self-training [17, 30] to adapt models to target domains. Similarly, the MTDA domain has witnessed significant attention [31, 30, 32, 29]. These studies tackle the challenge of adapting segmentation models to multiple target domains and employ diverse methodologies to achieve this objective. In contrast to UDA and MTDA, DG methods are characterized by their focus on learning domain-invariant representations. Researchers in this domain have explored techniques like learning domain invariant representation [33, 34, 35], and data augmentation[36, 37, 38], to impart models with generalization capabilities across various domains. Turning to the specific context of OCDA, studies have employed diverse strategies, including style regression based on scene-level attributes [8, 9, 10], and category-level attributes [7] to address the unique challenges posed by open compound domain adaptation for segmentation task.

The existing methods adapt scene styles to fill in the gap of image appearances between different domains. They are insensitive to the object styles, which are critical to capture the style change of the object appearances of various categories/instances. In contrast, we propose object-style compensation by respecting the individual categories/instances. Our approach appropriately adapts the object styles to the styles similar to the source domain and yield consistent context in the same scene.

### 2.2  Deep Network with Memorization for Visual Understanding

Recent studies demonstrate the importance of the deep network with memorization for visual understanding [39, 40, 41, 42, 43, 44, 45, 45, 46, 47, 48, 49, 50, 51, 52, 53, 54, 55]. Among them, the application of memorization techniques has been notably observed within the realms of image processing [50, 52, 55] and video analysis [44, 56]. Furthermore, these methods can be categorized based on the level of granularity at which objects are stored in memory, with some operating at the scene level [43, 50, 56], while others are geared towards the category level [52, 55].

The previous methods utilize memory that stores scene-level or category-level features extracted from the images. Because these features mainly capture a single domain's image style, they are less powerful for adapting a broad range of image styles of the compounded target domain to the source domain. In contrast, we propose the external memory to store both category- and instance-level discrepancy features learned across different domains. These discrepancy features can directly

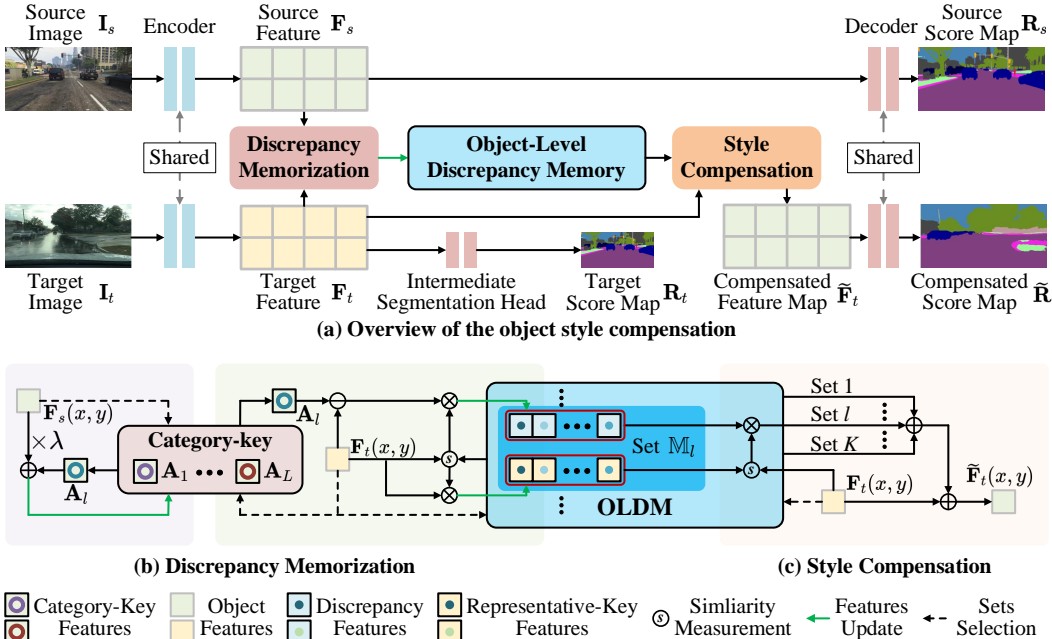

**(a) Overview of the object style compensation**

**(b) Discrepancy Memorization**  **(c) Style Compensation**

○ Category-Key  ☐ Object  ● Discrepancy  ● Representative-Key  ⓢ Simliarity  ← Features  ← Sets
○ Features  ☐ Features  ● Features  ● Features  Measurement  Update  Selection

Figure 2: (a) Overview of the object style compensation including discrepancy memorization, object-level discrepancy memory, and style compensation. (b) In the discrepancy memorization, we use the object features of the source and target domains, along with their differences, to update the category-key, representative-key, and discrepancy features in OLDM. (c) Given the object features of the target domain, we compute the similarities between the representative-key features and each object feature of the target domain, which is compensated by the discrepancy features weighted by the similarities.

compensate the images of the target domains, whose category- and instance-level styles are adapted to the source domain. The alignment of the object styles in different domains' images helps the segmentation network to focus on the intrinsic object appearances for producing the pseudo labels.

## 3 Method Overview

We illustrate the object style compensation in Figure 2(a). $\mathbf{I}_s, \mathbf{I}_t \in \mathbb{R}^{H \times W \times 3}$ are the images in the source and target domains. $\mathbf{I}_s/\mathbf{I}_t$ is associated with/without the pixel-wise annotation. We denote $\mathbf{Y}_s \in \mathbb{R}^{H \times W \times L}$ ($L$ is the category number) as the ground-truth annotation for $\mathbf{I}_s$. The encoder extracts the object feature maps $\mathbf{F}_s, \mathbf{F}_t \in \mathbb{R}^{H \times W \times C}$ from $\mathbf{I}_s, \mathbf{I}_t$. $C$ indicates the feature channels. $\mathbf{F}_s(x, y), \mathbf{F}_t(x, y) \in \mathbb{R}^C$ represent the object located at $(x, y)$ in the source and target images.

The object style compensation includes *Discrepancy Memorization* and *Style Compensation*. The discrepancy memorization uses $\mathbf{F}_s, \mathbf{F}_t$ to construct the *Object-Level Discrepancy Memory* (OLDM). OLDM stores the discrepancy features, which capture the change of object styles from target to source domain. For the target image $\mathbf{I}_t$, the style compensation uses discrepancy features to adapt the object style information of $\mathbf{F}_t$, yielding the compensated feature map $\widetilde{\mathbf{F}}_t \in \mathbb{R}^{H \times W \times C}$ used by decoder to compute the pseudo annotation $\mathbf{Y}_t \in \mathbb{R}^{H \times W \times L}$.

**Discrepancy Memorization**  The discrepancy memorization only takes place during network training. Here, we construct category-key and OLDM as $\{(\mathbf{A}_l, \mathbb{M}_l) \mid \mathbf{A}_l \in \mathbb{R}^C, l = 1, ..., L\}$, which contains pairs of category-key feature (e.g., $\mathbf{A}_l$) and feature set (e.g., $\mathbb{M}_l$). $\mathbf{A}_l$ is the category-key feature that captures the source domain's representative style of the $l^{th}$ category. The set $\mathbb{M}_l = \{(\mathbf{N}_{l,m}, \mathbf{D}_{l,m}) \mid \mathbf{N}_{l,m}, \mathbf{D}_{l,m} \in \mathbb{R}^C, m = 1, ..., M\}$ has pairs of representative-key feature (e.g., $\mathbf{N}_{l,m}$) and discrepancy feature (e.g., $\mathbf{D}_{l,m}$), where $m$ is the index of the representative-key feature. $\mathbf{N}_{l,m}$ captures a representative instance-level style of the $l^{th}$ category in target domain. We associate $\mathbf{N}_{l,m}$ with the discrepancy feature $\mathbf{D}_{l,m}$, which captures the change of the representative instance-level style from target to source domain.

We illustrate the discrepancy memorization in Figure 2(b), where we update the category-key, representative-key, and discrepancy features. We use the $l^{th}$ category's object features in $\mathbf{F}_s$ to update the category-key feature $\mathbf{A}_l$. In the set $\mathbb{M}_l$, we measure the similarities between the $l^{th}$ category's object features in $\mathbf{F}_t$ and the representative-key features $\{\mathbf{N}_{l,m} \mid m = 1, ..., M\}$. According to these similarities, we use the $l^{th}$ category's object feature $\mathbf{F}_t(x, y)$ updates representative-key features, while calculating the difference between $\mathbf{A}_l$ and $\mathbf{F}_t(x, y)$ to update the discrepancy features $\{\mathbf{D}_{l,m} \mid m = 1, ..., M\}$.

**Style Compensation** We conduct the style compensation during training and testing. We illustrate the style compensation in Figure 2(c), where we use the discrepancy features in OLDM to compensate the target image's object feature map $\mathbf{F}_t$. Given the object feature $\mathbf{F}_t(x, y)$ as a query, we use a segmentation head to regress its category scores $\mathbf{R}_t(x, y) \in \mathbb{R}^{H \times W \times L}$, selecting the feature sets $\{\mathbb{M}_1, ..., \mathbb{M}_K\}$ of OLDM for the style compensation. In $\mathbb{M}_k$, we calculate the similarities between $\mathbf{F}_t(x, y)$ and the representative-keys $\{\mathbf{N}_{k,m} \mid m = 1, ..., M\}$. These similarities weight the discrepancy features $\{\mathbf{D}_{k,m} \mid m = 1, ..., M\}$. After averaging the weighted discrepancy features of each set, the results are added to $\mathbf{F}_t(x, y)$, yielding the compensated feature $\widetilde{\mathbf{F}}_t(x, y)$. $\widetilde{\mathbf{F}}_t$ is fed into the decoder for regressing the pseudo annotation $\mathbf{Y}_t$. We borrow the pseudo annotation $\mathbf{Y}_t$ paired with the target image $\mathbf{I}_t$ to fine-tune the segmentation network.

# 4 Object Style Compensation

## 4.1 Discrepancy Memorization

During network training, we employ discrepancy memorization to construct the *Object-Level Discrepancy Memory* (OLDM). We denote category-key and OLDM as $\{(\mathbf{A}_l, \mathbb{M}_l) \mid \mathbf{A}_l \in \mathbb{R}^C, l = 1, ..., L\}$, where $\mathbb{M}_l = \{(\mathbf{N}_{l,m}, \mathbf{D}_{l,m}) \mid \mathbf{N}_{l,m}, \mathbf{D}_{l,m} \in \mathbb{R}^C, m = 1, ..., M\}$. We use the feature maps $\mathbf{F}_s, \mathbf{F}_t \in \mathbb{R}^{H \times W \times C}$, which are extracted from the source and target images $\mathbf{I}_s, \mathbf{I}_t \in \mathbb{R}^{H \times W \times 3}$ by encoder, to update the category-key, representative-key, and discrepancy features in OLDM (see Figure 2(b)).

**Update of Category-Key Features** Given the feature map $\mathbf{F}_s$ of the source image $\mathbf{I}_s$, we update the category-key features $\{\mathbf{A}_l \in \mathbb{R}^C \mid l = 1, ..., L\}$. We employ decoder to predict a category for every object feature in the map $\mathbf{F}_s$. The source image $\mathbf{I}_s$ has the ground-truth annotation $\mathbf{Y}_s \in \mathbb{R}^{H \times W \times L}$. The decoder predicts the category scores $\mathbf{R}_s(x, y) \in \mathbb{R}^L$, where $\mathbf{R}_s(x, y, l) \in \mathbb{R}$ is the score for predicting the object located at $(x, y)$ of the source image to be the $l^{th}$ category. We assume the category $l$ leads to the highest score $\mathbf{R}_s(x, y, l)$. The predicted label is compared to the ground-truth label $\mathbf{Y}_s(x, y)$. The correct prediction means the object feature $\mathbf{F}_s(x, y)$ is reliable for updating the category-key feature $\mathbf{A}_l$ as:

$$\mathbf{A}_l \leftarrow \mathbf{A}_l + \lambda \mathbf{F}_s(x, y),$$
$$s.t. \ \mathbf{R}_s(x, y, l) = \max\{\mathbf{R}_s(x, y, 1), ..., \mathbf{R}_s(x, y, L)\}, \quad l = \mathbf{Y}_s(x, y), \tag{1}$$

where $\leftarrow$ means the update by overwriting. $\lambda$ is a ratio for controlling the information of $\mathbf{F}_s(x, y)$ injected into $\mathbf{A}_l$. In Eq.(1), $\mathbf{A}_l$ aggregates the information of an array of object features, which consistently point to the $l^{th}$ category. It helps $\mathbf{A}_l$ to capture the representative object style of the $l^{th}$ category in source domain, which is used to compute discrepancy features.

**Update of Representative-Key and Discrepancy Features** We use the feature map $\mathbf{F}_t$ of the target image $\mathbf{I}_t$ to update the representative-key features in OLDM. For the object feature $\mathbf{F}_t(x, y) \in \mathbb{R}^C$, we use the intermediate segmentation head to regress the category scores in $\mathbf{R}_t(x, y) \in \mathbb{R}^L$, where intermediate segmentation head takes input as the target feature for regressing the category score map, and $\mathbf{R}_t(x, y, l) \in \mathbb{R}$ is the score for predicting the object located at $(x, y)$ of the target image to be the $l^{th}$ category. We use $\mathbf{F}_t(x, y)$ to update the representative-key feature $\mathbf{N}_{l,m}$ as:

$$\mathbf{N}_{l,m} \leftarrow \mathbf{N}_{l,m} + \mathbf{w}_{l,m} \mathbf{F}_t(x, y), \ \ \mathbf{D}_{l,m} \leftarrow \mathbf{D}_{l,m} + \mathbf{w}_{l,m}(\mathbf{A}_l - \mathbf{F}_t(x, y)),$$
$$s.t. \ \mathbf{R}_t(x, y, l) = \max\{\mathbf{R}_t(x, y, 1), ..., \mathbf{R}_t(x, y, L)\} > \gamma; \ \mathbf{w}_{l,m} = \frac{\mathbf{N}_{l,m} \cdot \mathbf{F}_t(x, y)}{\sqrt{C}}. \tag{2}$$

The category $l$ has the highest score in $\{\mathbf{R}_t(x, y, 1), ..., \mathbf{R}_t(x, y, L)\}$. We classify the object feature $\mathbf{F}_t(x, y)$ to the $l^{th}$ category, thus updating the representative-key features $\{\mathbf{N}_{l,m} \mid m = 1, ..., M\}$

in the set $\mathbb{M}_l$ for the same category. $\mathbf{w}_{l,m} \in \mathbb{R}$ is a weight, which is calculated as the dot-product between the representative-key feature $\mathbf{N}_{l,m}$ and the object feature $\mathbf{F}_t(x, y)$, whose similarity is measured. A high weight lets object feature contribute more information to representative-key feature.

Unlike decoder supervised by the ground-truth annotations of source images, the intermediate segmentation head is supervised by the pseudo annotations of target images. It may misclassify object feature, thus misleading the update of representative-key features. To remedy this issue, we set a score threshold $\gamma$ in Eq. (2), to select the reliable object feature $\mathbf{F}_t(x, y)$, whose highest score $\mathbf{R}_t(x, y, l)$ is greater than $\gamma$. This threshold only allows the object feature, whose category scores are confident, to update representative-key features. Each representative-key feature aggregates the features of various object instances with similar styles.

In Eq (2), we also update the discrepancy features $\{\mathbf{D}_{l,m} \in \mathbb{R}^C \mid m = 1, ..., M\}$. Here, we subtract the object feature $\mathbf{F}_t(x, y)$ from the category-key feature $\mathbf{A}_l$. Note that $\mathbf{F}_t(x, y)$ and $\mathbf{A}_l$ are extracted from the target and source images, respectively. Thus, the subtraction of these features captures the difference between the object styles of in target and source images, which is weighted by $\mathbf{w}_{l,m}$ to add to the discrepancy feature $\mathbf{D}_{l,m} \in \mathbb{R}^C$.

We remark that the update of OLDM is disabled during network testing. The style compensation uses the most updated category-key, representative-key, and discrepancy features in OLDM.

## 4.2 Style Compensation

The style compensation works during network training and testing. It harnesses the category- and representative-key features to find the appropriate discrepancy features in OLDM, which compensate for the object features of target image (see Figure 2(c)).

**Compensation of Object Features**  Given the object feature $\mathbf{F}_t(x, y)$ as a query, which is extracted from the target image $\mathbf{I}_t$, we use the intermediate segmentation head to predict the category scores $\mathbf{R}_t(x, y)$. Next, we select the representative-key and discrepancy features from $\{\mathbb{M}_1, ..., \mathbb{M}_K\}$, where $\mathbb{M}_k = \{(\mathbf{N}_{k,m}, \mathbf{D}_{k,m}) \mid \mathbf{N}_{k,m}, \mathbf{D}_{k,m} \in \mathbb{R}^C, m = 1, ..., M\}$ to compensate the object feature $\mathbf{F}_t(x, y)$, yielding the compensated feature $\widetilde{\mathbf{F}}_t(x, y) \in \mathbb{R}^C$ as:

$$\widetilde{\mathbf{F}}_t(x, y) = \mathbf{F}_t(x, y) + \sum_k \sum_m \mathbf{w}_{k,m} \mathbf{D}_{k,m},$$

$$s.t. \ \mathbf{R}_t(x, y, k) \in \max{}_K \{\mathbf{R}_t(x, y, 1), ..., \mathbf{R}_t(x, y, L)\}; \ \mathbf{w}_{k,m} = \frac{\mathbf{N}_{k,m} \cdot \mathbf{F}_t(x, y)}{\sqrt{C}}, \quad (3)$$

where $\max_K$ means to select the top-K relevant categories. We use Eq. (3) to achieve the compensated feature $\widetilde{\mathbf{F}}_t \in \mathbb{R}^{H \times W \times C}$, which is used for computing the pseudo annotation for the target image $\mathbf{I}_t$.

**Computation of Pseudo Annotations**  Based on the compensated feature $\widetilde{\mathbf{F}}_t$, the decoder of the segmentation network predicts the category score map $\widetilde{\mathbf{R}}_t \in \mathbb{R}^{H \times W \times L}$. We use the category score map $\widetilde{\mathbf{R}}_t$ to produce the pseudo annotation $\mathbf{Y}_t \in \mathbb{R}^{H \times W \times L}$, where the pixel-wise annotation $\mathbf{Y}_t(x, y) \in \mathbb{R}^L$ for the pixel located at $(x, y)$ is determined as:

$$\mathbf{Y}_t(x, y, l) = \begin{cases} 1 & \max\{\widetilde{\mathbf{R}}_t(x, y, 1), ..., \widetilde{\mathbf{R}}_t(x, y, L)\} = \widetilde{\mathbf{R}}_t(x, y, l) > \gamma, \\ 0 & \max\{\widetilde{\mathbf{R}}_t(x, y, 1), ..., \widetilde{\mathbf{R}}_t(x, y, L)\} > \max\{\widehat{\mathbf{R}}_t(x, y, l), \gamma\}, \\ \text{ignored} & \gamma \geq \max\{\widetilde{\mathbf{R}}_t(x, y, 1), ..., \widetilde{\mathbf{R}}_t(x, y, L)\}. \end{cases} \quad (4)$$

The pixel-wise annotation $\mathbf{Y}_t(x, y)$ is a one-hot vector. We set the $l^{th}$ channel $\mathbf{Y}_t(x, y, l)$ to 1 (see the first case), when the category score $\widetilde{\mathbf{R}}_t(x, y, l)$ is higher than other scores in the set $\{\widetilde{\mathbf{R}}_t(x, y, 1), ..., \widetilde{\mathbf{R}}_t(x, y, L)\}$; otherwise, we set $\mathbf{Y}_t(x, y, l)$ to 0 (see the second case). It should be noted that the threshold $\gamma$ is used in the third case, where too low scores let the pseudo annotations be ignored during network training. The computation of pseudo annotations only takes place during network training. For network testing, we resort to the category score map $\widetilde{\mathbf{R}}_t$ to predict the labels for all pixels, following the convention of semantic segmentation.

## 4.3 Supervision for Network Training

We use the ground-truth annotations of source images and the pseudo annotations of target images to train segmentation network. We formulate the overall training objective $\mathcal{L}$ as:

$$\mathcal{L} = \mathcal{L}_{gt} + \mathcal{L}_{pse}. \tag{5}$$

$\mathcal{L}_{gt}$ and $\mathcal{L}_{pse}$ represent the objectives with the supervision of ground-truth and pseudo annotations.

**Supervision of Ground-truth Annotations**  Based on the object feature map $\mathbf{F}_s$ of the source image $\mathbf{I}_s$, decoder predicts the category score map $\mathbf{R}_s$. We use the cross-entropy loss $\mathcal{CE}$ to penalize the difference between the predicted score map $\mathbf{R}_s$ and the ground-truth annotation $\mathbf{Y}_s$ as:

$$\mathcal{L}_{gt} = \mathcal{CE}(\mathbf{R}_s, \mathbf{Y}_s). \tag{6}$$

**Supervision of Pseudo annotations**  We use intermediate segmentation head to regress the category score map $\mathbf{R}_t$ for the target image $\mathbf{I}_t$, based on the object feature map $\mathbf{F}_t$. We compute the cross-entropy loss, which penalizes the difference between the score map $\mathbf{R}_t$ and the pseudo annotation $\mathbf{Y}_t$ as the first term of the below Eq. (7).

$$\mathcal{L}_{pse} = \mathcal{CE}(\mathbf{R}_t, \mathbf{Y}_t) + \mathcal{CE}(\widetilde{\mathbf{R}}_t, \mathbf{Y}_t). \tag{7}$$

Moreover, we leverage the decoder to predict the category score map $\widetilde{\mathbf{R}}_t$, based on the compensated feature map $\widetilde{\mathbf{F}}_t$. We again use cross-entropy loss (see the second term of Eq. (7)) to measure the segmentation errors in the score map $\widetilde{\mathbf{R}}_t$, by comparing to the pseudo annotation $\mathbf{Y}_t$.

# 5 Experiments

## 5.1 Experimental Datasets

We use GTA5 [57], SYNTHIA [58], C-Driving [7], ACDC [12], Cityscapes [13], KITTI [14], and WildDash [15] datasets to evaluate our method. The images in a dataset may be subdivided into source, target, and open domains. All images with annotations in source domain and a portion of images without annotations in target domain are used for network training. We evaluate the segmentation performances on the rest images in target domain and all images in open domain. We list the division of these datasets in Table 1.

Table 1: Divisions of the experimental datasets.

| Dataset | Total | Train | | Test | |
|---|---|---|---|---|---|
| | | Source | Target | | Open |
| GTA5 | 24,966 | 24,966 | - | - | - |
| SYNTHIA | 9,400 | 9,400 | - | - | - |
| C-Driving | 16,127 | - | 14,697 | 803 | 627 |
| ACDC | 1,906 | - | 1,200 | 306 | 400 |
| Cityscapes | 500 | - | - | - | 500 |
| KITTI | 200 | - | - | - | 200 |
| WashDash | 638 | - | - | - | 638 |

In the alation study of our method, we use 24,966 source images of GTA5 dataset and 14,697 target images of C-Driving dataset for network training. 803 and 627 images of target and open domains in C-Driving dataset are used for testing. We report the segmentation performance regarding mean intersection-over-unions (mIoUs) on target and open domains.

## 5.2 Implemenration Detail

We employ the PyTorch toolkit to implement our segmentation network with *Object-Level Discrepancy Memory* (OLDM). Our segmentation architecture is built upon DeepLab-V2 [59] with ResNet101 [60] backbone that has been pre-trained on ImageNet [61]. To optimize the network, we utilize the SGD solver for 250,000 iterations. The initial learning rate is set to 0.00025, which undergoes a linear decay throughout the training process. Each mini-batch consists of 4 images, comprising 2 source images and 2 target images. We apply random cropping, flipping, color jittering, and Gaussian blurring techniques to the training images, using a crop size of $640 \times 360$. The network is trained on a single RTX 3090 GPU.

## 5.3 Ablation Study

**Analysis of Discrepancy Memorization**  In Tables 2, 3, and 4, we study various strategies for updating category-key, representative-key, and discrepancy features during the discrepancy memorization.

We report different strategies for using category-key features in Table 2. First, we disable the update of category-key features. This is done by removing OLDM (see "w/o OLDM") during training and testing, producing the performance of the stand-alone segmentation network. Another alternative uses the mean of the representative-key features in the same set to replace category-key feature but yields lower performances than our method (see "mean instances"). This is be-

Table 2: Results of various ways of using category-key features. mIoU(T) and mIoU(O) mean the mIoUs on target and open domains.

| Category-Key | Method | mIoU(T) | mIoU(O) |
|---|---|---|---|
| ✘ | w/o OLDM | 36.6 | 39.7 |
| | mean instances | 39.2 | 41.5 |
| ✔ | local | 41.7 | 43.2 |
| | global | **44.1** | **46.9** |

cause category-key and instance features are computed based on the image features of discrepant domains (i.e., source and target domains), making none of them replaceable. Second, we experiment with using the mean of the image features of source domain in a local mini-batch to compute the category-key features, which are overridden by new mini-batch. This strategy achieves worse results than our method (see "local"), because we use different mini-batches to compute category-key features globally (see "global"), which more comprehensively capture the object features of source domain.

In Table 3, we compare the performances of various strategies for using representative-key features. By eliminating the representative-key features in OLDM, we lack the similarities between the object features of target domain and the representative-key features for weighting the discrepancy features to compensate for object features. In this case, we experiment with simply averaging discrepancy features (see "mean discrepancy"), using the category-level discrepancy rather than the instance-level, or directly

Table 3: Results of various ways of using representative-key features.

| Rep.-Key | Method | mIoU(T) | mIoU(O) |
|---|---|---|---|
| ✘ | mean discrepancy | 37.8 | 38.8 |
| | category-level discrepancy | 38.1 | 40.2 |
| | discrepancy similarity | 39.1 | 40.2 |
| ✔ | top-1 update | 40.7 | 41.8 |
| | top-50% update | 42.2 | 44.3 |
| | 100% update | **44.1** | **46.9** |

computing the similarities between object features and discrepancy features (see "discrepancy similarity") for compensation. Without representative-key features, these naive strategies lead to worse results than our method. This is because the average discrepancy features and the alternative similarities inappropriately match object instances to the discrepancy features desired for compensation.

We also experiment with enabling and updating representative-key features in various ways. For each set of representative-key and discrepancy features of the same category, we select the representative-key feature, which is updated along with associated discrepancy feature by the object feature of target domain. It degrades the performances (see "top-1 update"), compared to adequately updating 50% or 100% of representative-key and discrepancy features (see "top-50% update" and "100% update").

In Table 4, we evaluate the performances of using discrepancy features in different ways. Removing all discrepancy features in OLDM, we again degrade the whole pipeline to the backbone segmentation network (see "w/o OLDM"). We also experiment with replacing discrepancy features with the discrepancy between the category- and representative-key features. This discrepancy only considers the difference between representative features of source and tar-

Table 4: Results of various ways of using discrepancy features.

| Discrepancy | Method | mIoU(T) | mIoU(O) |
|---|---|---|---|
| ✘ | w/o OLDM | 36.6 | 39.7 |
| | key discrepancy | 42.7 | 44.3 |
| ✔ | merged sets | 39.6 | 41.5 |
| | multi-sets, k-means | 41.7 | 43.3 |
| | multi-sets, category | **44.1** | **46.9** |

get domains. Our discrepancy features account for the differences between the representative features of source domain and various instances' object features of target domain. They show a stronger power for compensating the style information of relevant object instances. Moreover, we evaluate several alternatives for enabling discrepancy features for compensation. In contrast to our method that differentiates discrepancy features according to categories and instances, an alternative method merges all discrepancy features as a set (see "merged sets"). This method is insensitive to the specific properties of categories and instances. Another way of separating discrepancy features into multiple sets is to use k-means clustering (see "multi-sets, k-means"), without depending on category-key features. However, these methods neglect the useful category-key features, which select

representative-key and discrepancy features for compensating the object features of the matched categories. Their performances lag behind our method (see "multi-sets, category").

**Variants of Style Compensation**   In Tables 5 and 6, we evaluate the effectiveness of the style compensation by replacing it with other schemes during network testing. Here, OLDM has the optimized category-key, representative-key, and discrepancy features.

First, we remove the optimized OLDM and disable the style compensation. Another alternative method of style compensation is averaging all of the discrepancy features without relying on the similarities between target images' and representative-key features for weighting discrepancy features. The above methods (see "w/o OLDM" and "mean discrepancy") yield lower performances than our method, which better utilizes discrepancy features for compensating the style information of different categories and instances in target domain.

Table 5: Results of various compensation methods.

| Style Compensation | Method | mIoU(T) | mIoU(O) |
|---|---|---|---|
| ✗ | w/o OLDM | 36.6 | 39.7 |
| ✔ | mean discrepancy | 40.7 | 41.9 |
| | instance similarity | **44.1** | **46.9** |

Next, we analyze the quality of the pseudo annotations computed by the style compensation. Without pseudo annotations, we only use the ground-truth annotations of source images for training the segmentation network. We also evaluate the quality of the pseudo annotations produced by the intermediate segmentation head for supervising the segmentation network. Without the accurate pseudo annotations computed based on compensated object features, these alternatives yield worse results than our method.

Table 6: Results of various ways of computing pseudo annotations.

| Pseudo Annotations | Method | mIoU(T) | mIoU(O) |
|---|---|---|---|
| ✗ | w/o pseudo | 39.7 | 41.6 |
| ✔ | intermediate | 42.4 | 44.3 |
| | final | **44.1** | **46.9** |

## 5.4   External Comparison

In Table 7 and 8, we compare the performances of start-of-the-art OCDA , UDA and DG methods [7, 8, 9, 10, 11, 62]. For a fair comparison, we have clearly specified the sets of images used for training and testing in each table. Our method of object style compensation outperforms an array of start-of-the-art methods on different datasets. In Figures 3 and 4, we compare the segmentation results of different methods, where our method yields better results.

## 6   Conclusion

Open compound domain adaptation has been successfully used to improve semantic image segmentation performance. The popular methods globally adapt the scene styles of images. However, they unreasonably change the object styles of various categories and instances, forming unusual object

Table 7: Comparison with state-of-the-art methods. We clarify the source, target, and open domains for training and testing the compared methods at the top of table. $mIoU^{11}$ and $mIoU^{16}$ mean the mIoUs on 11 and 16 categories, respectively.

| Method | Type | Train:C-Driving(T). Test:C-Driving(T) | | | Train:ACDC(T). Test: ACDC(T & O) | | | |
|---|---|---|---|---|---|---|---|---|
| | | GTA5(S) | SYNTHIA(S) | | GTA5(S) | | SYNTHIA(S) | |
| | | mIoU(T) | $mIoU^{16}$(T) | $mIoU^{11}$(T) | mIoU(T) | mIoU(O) | $mIoU^{16}$(T) | $mIoU^{16}$(O) |
| Source-only | - | 28.3 | 20.9 | 28.1 | 20.5 | 27.1 | 19.8 | 20.5 |
| CDAS[7] | OCDA | 31.4 | 25.3 | 34.0 | 25.3 | 29.1 | 25.9 | 23.3 |
| CSFU[9] | OCDA | 34.9 | 26.1 | 34.8 | 27.6 | 30.5 | 26.7 | 24.8 |
| DACS[62] | UDA | 36.6 | 28.1 | 36.5 | 29.0 | 34.8 | 28.3 | 27.0 |
| DHA[8] | OCDA | 37.1 | 29.9 | 37.6 | 29.5 | 37.5 | 29.2 | 27.3 |
| AST[10] | OCDA | 38.8 | 31.1 | 38.9 | 30.7 | 39.2 | 30.1 | 27.9 |
| ML-BPM[11] | OCDA | 40.2 | 32.1 | 40.0 | 32.1 | 41.6 | 31.9 | 29.1 |
| Ours | OCDA | **44.1** | **35.6** | **43.7** | **35.7** | **44.1** | **34.7** | **36.4** |

Table 8: Comparison with state-of-the-art methods. CD, CS, KT, and WD mean the mIoUs on the C-Driving, Cityscapes, KITTI, and WildDash datasets.

| Method | Type | GTA5(S) | | | | | SYNTHIA(S) | | | | |
|---|---|---|---|---|---|---|---|---|---|---|---|
| | | CD | CS | KT | WD | Avg | CD | CS | KT | WD | Avg |
| CSFU[9] | OCDA | 38.9 | 38.6 | 37.9 | 29.1 | 36.1 | 36.2 | 34.9 | 32.4 | 27.6 | 32.8 |
| DACS[62] | UDA | 39.7 | 37.0 | 40.2 | 30.7 | 36.9 | 36.8 | 37.0 | 37.4 | 28.8 | 35.0 |
| RobustNet[63] | DG | 38.1 | 38.3 | 40.5 | 30.8 | 37.0 | 37.1 | 38.3 | 40.1 | 29.6 | 36.3 |
| DHA[8] | OCDA | 39.4 | 38.8 | 40.1 | 30.9 | 37.5 | 38.9 | 38.0 | 40.6 | 30.0 | 36.9 |
| AST[10] | OCDA | 40.7 | 40.3 | 41.9 | 32.2 | 38.8 | 40.5 | 39.8 | 41.6 | 30.7 | 38.2 |
| ML-BPM[11] | OCDA | 42.5 | 41.7 | 44.3 | 34.6 | 40.8 | 42.6 | 41.1 | 43.4 | 30.9 | 39.5 |
| **Ours** | OCDA | **46.9** | **43.6** | **46.5** | **40.1** | **44.3** | **48.5** | **48.0** | **51.3** | **39.6** | **46.9** |

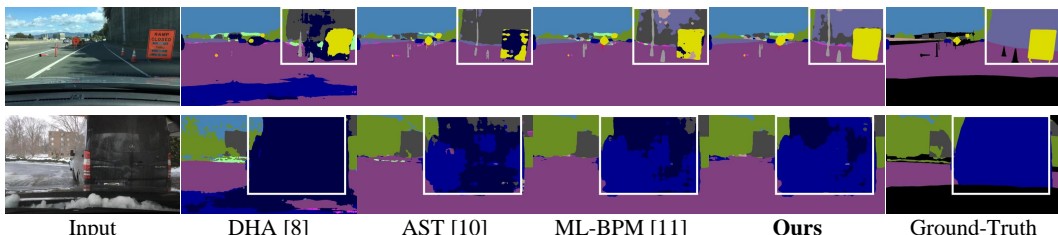

| Input | DHA [8] | AST [10] | ML-BPM [11] | **Ours** | Ground-Truth |

Figure 3: Segmentation results of different methods on the target domain of C-Driving.

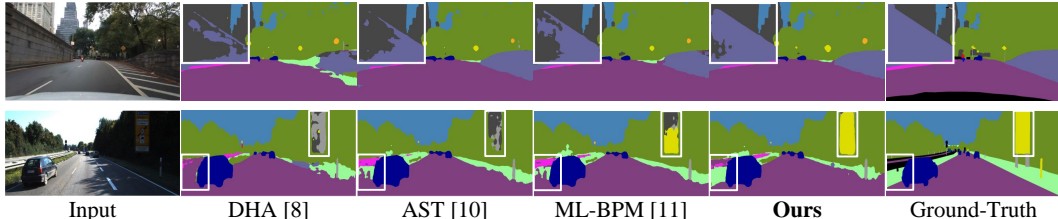

| Input | DHA [8] | AST [10] | ML-BPM [11] | **Ours** | Ground-Truth |

Figure 4: Segmentation results of various methods on the open domains of C-Driving and KITTI.

contexts in the adapted images along with incorrect pseudo annotations. In this paper, we propose a novel idea of object-style compensation. Our method leverages the object-level discrepancy memory, where multiple sets of discrepancy features account for the style changes of objects in different categories. Furthermore, the discrepancy features in the same set individually consider instance-level style changes, thus adapting the object styles finely. Our method enables a more accurate computation of the pseudo annotations of the images in the target domains, which eventually assists in training the segmentation network. In the future, we will extend our method to other applications (e.g., object detection and image restoration).

## Negative Societal Impacts

Our approach facilitates the analysis of image semantics, rendering it applicable in various scenarios such as autonomous vehicles and image/video content comprehension. However, it is important to exercise caution when utilizing the results, as they may contain potentially problematic information. Mishandling such information could lead to infringements on privacy or economic interests.

## Acknowledgments

We thank the anonymous reviewers for their constructive suggestions. This work is supported in parts by the National Key Research and Development Program of China (2020YFC1522701) and the National Natural Science Foundation of China (62072334).

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
