# Open Compound Domain Adaptation with Object Style Compensation for Semantic Segmentation
# – Supplementary Material –

**Tingliang Feng**[1,2*]     **Hao Shi**[1,3*]     **Xueyang Liu**[1]     **Wei Feng**[1,2]
**Liang Wan**[1]     **Yanlin Zhou**[4]     **Di Lin**[1†]

[1]College of Intelligence and Computing, Tianjin University
[2]Laboratory of Computation and Analytics of Complex Management Systems (CACMS), Tianjin University
[3]Department of Automation, Tsinghua University     [4]Dunhuang Academy
{fengtl, xyliu850569498, lwan}@tju.edu.cn     shi-h23@mails.tsinghua.edu.cn
wfeng@ieee.org     zhouyanlin@dha.ac.cn     Ande.lin1988@gmain.com

## 1 Implementation Details

In our implementation, we adopt a warm-up strategy to pretrain the backbone network for 50,000 iterations. After the warm-up phase, we generate pseudo annotations for the training data in target domain and determine the initial values of category-key features based on source images. Next, we initialize the representative-key features and discrepancy features of each set in OLDM using a First-Input, Fist-Output (FIFO) queue. To ensure comprehensive initialization of all sets in OLDM, this phase extends over 4000 iterations. Upon completion, our method empowers the execution of Discrepancy Memorization and Style Compensation.

## 2 Supplementary Experiments

In this section, we provide an extensive and comprehensive display of experiments. We have structured this section into five subsections. The initial subsection provides an in-depth analysis of the state-of-the-art comparison experiments outlined in the main paper. The subsequent subsection focuses on conducting a sensitivity analysis of the hyperparameters mentioned in the main paper. Following that, we evaluate OLDM in cross-model scenarios as the third subsection. The fourth subsection delves into the analysis of the compensation effect. Lastly, the fifth subsection showcases a broader range of visual results obtained from different datasets.

### 2.1 Comprehensive Experimental Results

In this section, we present a comprehensive experimental results of state-of-the-art methods in Table 1, 2, 3 and 4, which is related to the Table 7 in the main paper.

Table 1: Comparison with the state-of-the-art methods. Train: GTA5(Source), C-Driving(Target). Test:C-Driving(Target). Related to the Table 7 (a) in the main paper.

| Method | Type | road | sidewalk | building | wall | fence | pole | light | sign | veg | terrain | sky | person | rider | car | truck | bus | train | mbike | bike | mIoU(T) |
|---|---|---|---|---|---|---|---|---|---|---|---|---|---|---|---|---|---|---|---|---|---|
| Source-only | - | 73.4 | 12.5 | 62.8 | 6.0 | 15.8 | 19.4 | 10.9 | 21.1 | 54.6 | 13.9 | 76.7 | 34.5 | 12.4 | 68.1 | 31.0 | 12.8 | 0.0 | 10.1 | 1.9 | 28.3 |
| CDAS[4] | OCDA | 79.1 | 9.4 | 67.2 | 12.3 | 15.0 | 20.1 | 14.8 | 23.8 | 65.0 | 22.9 | 82.6 | 40.4 | 7.2 | 73.0 | 27.1 | 18.3 | 0.0 | 16.1 | 1.5 | 31.4 |
| CSFU[5] | OCDA | 80.1 | 12.2 | 70.8 | 9.4 | 24.5 | 22.8 | 19.1 | 30.3 | 68.5 | 28.9 | 82.7 | 47.0 | 16.4 | 79.9 | 36.6 | 18.8 | 0.0 | 13.5 | 1.4 | 34.9 |
| DACS[6] | UDA | 81.9 | 24.0 | 72.2 | 11.9 | 28.6 | 24.2 | 18.3 | 35.4 | 71.8 | 28.0 | 87.7 | 44.9 | 15.6 | 78.4 | 39.1 | 24.9 | 0.1 | 6.9 | 1.9 | 36.6 |
| DHA[7] | OCDA | 79.9 | 14.5 | 71.4 | 13.1 | 32.0 | 27.1 | 20.7 | 35.3 | 70.5 | 27.5 | 86.4 | 47.3 | 23.3 | 77.6 | 44.0 | 18.0 | 0.1 | 13.7 | 2.5 | 37.1 |
| AST[8] | OCDA | 82.3 | 20.3 | 70.7 | 11.2 | 31.8 | 25.8 | 23.1 | 37.5 | 72.5 | 26.5 | 86.3 | 48.1 | 30.1 | 78.4 | 48.1 | 25.6 | 0.0 | 18.2 | 1.1 | 38.8 |
| ML-BPM[9] | OCDA | 85.3 | **26.2** | 72.8 | 10.6 | 33.1 | 26.9 | **24.6** | 39.4 | 70.8 | 32.5 | 87.9 | 47.6 | 29.2 | **84.8** | 46.0 | 22.8 | **0.2** | 16.7 | **5.8** | 40.2 |
| **Ours** | OCDA | **87.3** | 23.5 | **76.0** | **27.9** | **35.9** | **29.3** | 22.3 | **43.8** | **75.6** | **39.0** | **90.3** | **55.5** | **40.6** | 84.0 | **55.0** | **30.5** | 0.0 | **21.8** | 0.0 | **44.1** |

---

*Co-first authors. Equal contribution.

†Di Lin is the corresponding author of this paper.

Table 2: Comparison with the state-of-the-art methods. Train: SYNTHIA(Source), C-Driving(Target). Test:C-Driving(Target). Related to the Table 7 (b) in the main paper.

| Method | Type | road | sidewalk | building | wall | fence | pole | light | sign* | veg | sky | person | rider* | car | bus* | mbike* | bike* | mIoU[16] | mIoU[11] |
|---|---|---|---|---|---|---|---|---|---|---|---|---|---|---|---|---|---|---|---|
| Source-only | - | 33.9 | 11.9 | 42.5 | 1.5 | 0.0 | 14.7 | 0.0 | 1.3 | 56.8 | 76.5 | 13.3 | 7.4 | 57.8 | 12.5 | 2.1 | 1.6 | 20.9 | 28.1 |
| CDAS[4] | OCDA | 54.5 | 13.0 | 53.9 | 0.8 | 0.0 | 18.2 | 13.0 | 13.2 | 60.0 | 78.9 | 17.6 | 3.1 | 64.2 | 12.2 | 2.1 | 1.5 | 25.3 | 34.0 |
| CSFU[5] | OCDA | 69.6 | 12.2 | 50.9 | 1.3 | 0.0 | 16.7 | 12.1 | 13.6 | 56.2 | 75.8 | 20.0 | 4.8 | 68.2 | 14.1 | 0.9 | 1.2 | 26.1 | 34.8 |
| DACS[6] | UDA | 62.1 | **15.2** | 48.8 | 0.3 | 0.0 | 19.7 | 10.3 | 9.6 | 57.8 | 84.4 | 35.2 | 18.9 | 67.8 | 16.0 | 2.2 | 1.7 | 28.1 | 36.5 |
| DHA[7] | OCDA | 67.5 | 2.5 | 54.6 | 0.2 | 0.0 | 25.8 | 13.4 | 27.1 | 58.0 | 83.9 | 36.0 | 6.1 | 71.6 | 28.9 | 2.2 | 1.8 | 29.9 | 37.6 |
| AST[8] | OCDA | 69.2 | 13.6 | 60.4 | 0.6 | 0.0 | 23.7 | 12.1 | 25.9 | 60.3 | 82.1 | 38.4 | 14.4 | 67.3 | 25.1 | 1.6 | 3.1 | 31.1 | 38.9 |
| ML-BPM[9] | OCDA | 73.4 | **15.2** | 57.1 | **1.8** | 0.0 | 23.2 | **13.5** | 23.9 | 59.9 | 83.3 | 40.3 | **22.3** | 72.2 | 23.3 | **2.3** | 2.2 | 32.1 | 40.0 |
| **Ours** | OCDA | **80.8** | 4.6 | **73.4** | 0.0 | 0.0 | **26.4** | 11.5 | **37.9** | **68.2** | **88.4** | **50.6** | 11.0 | **76.4** | **33.1** | 1.5 | **6.2** | **35.6** | **43.7** |

Table 3: Comparison with the state-of-the-art methods. Train: GTA5(Source), ACDC(Target). Test:ACDC(Target). Related to the Table 7 (e) in the main paper.

| Method | Type | road | sidewalk | building | wall | fence | pole | light | sign | veg | terrain | sky | person | rider | car | truck | bus | train | mbike | bike | mIoU(T) |
|---|---|---|---|---|---|---|---|---|---|---|---|---|---|---|---|---|---|---|---|---|---|
| Source-only | - | 43.6 | 2.5 | 46.2 | 5.2 | 0.1 | 30.3 | 15.3 | 16.3 | 56.9 | 0.0 | 71.5 | 16.3 | 13.7 | 51.4 | 0.0 | 15.1 | 0.0 | 1.4 | 4.2 | 20.5 |
| CDAS[4] | OCDA | 53.2 | 5.9 | 56.1 | 10.1 | 2.6 | 22.0 | 37.1 | 11.4 | 53.9 | 23.5 | 71.3 | 27.6 | 14.6 | 47.5 | 16.8 | 19.5 | 0.0 | 3.2 | 3.8 | 25.3 |
| CSFU[5] | OCDA | 47.0 | 4.1 | 53.0 | 13.9 | 1.0 | 23.2 | 41.2 | 18.8 | 55.8 | 23.2 | 72.1 | 31.5 | 10.8 | 69.1 | 26.4 | 27.8 | 0.2 | 1.7 | 2.6 | 27.6 |
| DACS[6] | UDA | 48.9 | 9.7 | 54.5 | 16.8 | 5.7 | 22.7 | 42.0 | 22.9 | 61.3 | 29.7 | 73.7 | 32.2 | 11.6 | 63.3 | 23.2 | 26.5 | 0.0 | 1.2 | 5.2 | 29.0 |
| DHA[7] | OCDA | 49.8 | 5.2 | 59.1 | 10.2 | 3.1 | 25.6 | 47.8 | 27.9 | 65.1 | 32.0 | 75.2 | 29.0 | 12.2 | 61.5 | 20.5 | 32.4 | 0.0 | 1.0 | 2.0 | 29.5 |
| AST[8] | OCDA | 51.3 | 7.2 | **60.2** | 19.3 | 8.6 | 29.8 | 48.6 | 18.5 | 58.3 | 29.8 | 74.8 | **35.3** | 9.5 | 70.3 | 28.4 | 24.8 | 3.1 | 2.7 | 3.2 | 30.7 |
| ML-BPM[9] | OCDA | 48.4 | 5.0 | 58.2 | 25.3 | 10.0 | **35.1** | **50.4** | **26.7** | **66.8** | 33.3 | **75.8** | 32.1 | **16.7** | **73.5** | 16.8 | 26.6 | 0.2 | 3.9 | **4.6** | 32.1 |
| **Ours** | OCDA | **72.1** | **18.7** | 50.9 | **26.5** | **14.3** | 34.5 | 46.4 | 11.4 | 61.4 | **34.7** | 72.5 | 31.5 | 5.6 | 72.1 | **44.2** | **65.1** | **10.5** | **5.2** | 0.2 | **35.7** |

Table 4: Comparison with the state-of-the-art methods. Train: SYNTHIA(Source), ACDC(Target). Test:ACDC(Target). Related to the Table 7 (f) in the main paper.

| Method | Type | road | sidewalk | building | wall | fence | pole | light | sign | veg | sky | person | rider | car | bus | mbike | bike | mIoU[16] |
|---|---|---|---|---|---|---|---|---|---|---|---|---|---|---|---|---|---|---|
| Source-only | - | 45.2 | 0.2 | 36.7 | 1.7 | 0.6 | 25.7 | 4.0 | 5.6 | 46.6 | 64.3 | 16.9 | 11.3 | 39.6 | 16.5 | 0.6 | 1.9 | 19.8 |
| CDAS[4] | OCDA | 61.3 | 0.7 | 60.1 | 11.7 | 1.8 | 28.4 | 18.8 | 23.5 | 48.6 | 28.9 | 16.5 | 15.9 | 69.2 | 18.4 | 5.4 | 5.6 | 25.9 |
| CSFU[5] | OCDA | 62.6 | 0.3 | 60.3 | 8.6 | 1.8 | 21.3 | 20.7 | 29.1 | 44.5 | 22.1 | 34.5 | 19.0 | 71.1 | 23.2 | 4.4 | 4.3 | 26.7 |
| DACS[6] | UDA | 55.6 | 1.1 | 55.7 | 0.1 | 0.7 | 25.8 | **31.7** | 18.3 | 65.5 | 53.7 | 31.1 | 16.6 | 69.2 | 22.5 | 2.9 | 3.1 | 28.3 |
| DHA[7] | OCDA | 55.5 | 1.1 | 57.2 | 0.7 | 0.8 | 26.6 | 22.7 | 24.6 | 65.8 | 58.4 | 29.6 | 23.9 | 70.8 | 19.5 | 5.4 | 4.2 | 29.2 |
| AST[8] | OCDA | 60.1 | 1.3 | 60.3 | 5.3 | 0.2 | 25.4 | 27.7 | 18.6 | 63.9 | 67.2 | 30.2 | 25.1 | 70.1 | 20.3 | 4.3 | 2.2 | 30.1 |
| ML-BPM[9] | OCDA | 66.7 | 1.7 | 62.4 | 10.8 | 1.4 | **30.8** | 23.9 | 29.2 | 62.6 | **69.0** | 31.6 | 14.6 | **71.8** | 22.9 | **6.8** | 4.5 | 31.9 |
| **Ours** | OCDA | **75.8** | **5.2** | **64.9** | **13.5** | **4.2** | 28.6 | 25.1 | 31.1 | **66.2** | 59.4 | **35.1** | **29.5** | 69.5 | **36.2** | 4.2 | **6.2** | **34.7** |

## 2.2 Sensitivity Analysis of Hyper-parameters

We assess the impact of key hyperparameters on the segmentation performance, namely the memory capacity $M$, the number of feature sets $K$, and the values of $\lambda$ and $\gamma$. These hyperparameters govern the capacity and update process of the Object-Level Discrepancy Memory (OLDM), as well as the management of category-key, representative-key, and discrepancy features during the discrepancy memorization phase.

**Sensitivity Analysis of Memory Capacity** We examine the influence on segmentation performance by varying the number of discrepancy features in each set of OLDM (see Figure 1). We select feature numbers from $\{10, 20, 30, 40, 50, 60, 70, 80\}$. As illustrated in Figure 1(a–b), insufficient discrepancy features fail to adequately capture style changes in instances adapted from the target to source domains, resulting in suboptimal performance. Conversely, an excess of discrepancy features proves redundant and saturates performance. Moreover, it demands substantial computational resources to search for appropriate discrepancy features from OLDM to compensate for style-related features (see Figure 1(c–d)). By default, we set $M = 50$.

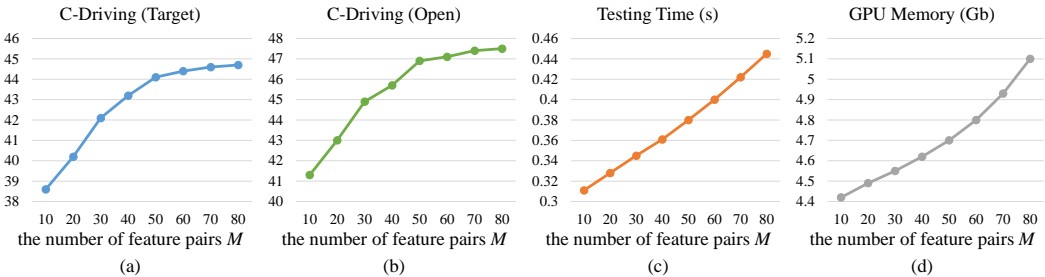

Figure 1: The results of sensitivity analysis of memory capacity on the target and open domain of C-Driving.

**Sensitivity Analysis of ratio $\lambda$** When updating the category-key features, $\lambda$ represents a ratio that controls the information injection of $\mathbf{F}_s(x, y)$ into $\mathbf{A}_l$ in Eq.(1) of the main paper. As depicted in Figure 2, we compare the segmentation performance using different values of $\lambda$. Setting $\lambda$ excessively large (e.g., $\lambda = 0.1, 0.05, 0.01$) renders the category-key features susceptible to the influence of subsequent features, compromising performance. Conversely, an excessively small $\lambda$ (e.g., $\lambda = 0.0005, 0.0001$) results in sluggish category-key feature updates, leading to high sensitivity to the initial value. By default, we use $\lambda = 0.001$.

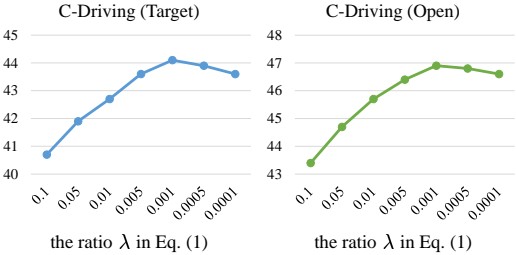

Figure 2: The results of sensitivity analysis of ratio $\lambda$ on the target and open domain of C-Driving.

**Sensitivity Analysis of score threshold $\gamma$** When updating the representative-key and discrepancy features, we introduce a score threshold $\gamma$ in Eq.(2) of the main paper to select reliable object features. In Figure 3, we investigate the effectiveness of different threshold settings. Setting $\gamma$ too small (e.g., $\gamma = 0, 0.1, 0.2, 0.3$) introduces more noise into the object features used for updating the representative-key and discrepancy features, hindering the effective construction of OLDM. Conversely, setting $\gamma$ too large (e.g., $\gamma = 0.7, 0.8, 0.9$) imposes overly stringent selection

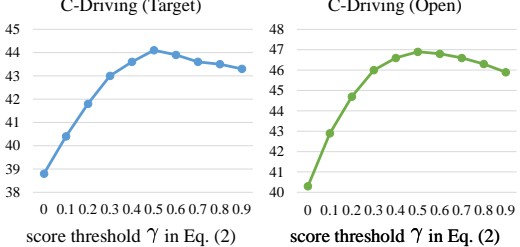

Figure 3: The results of sensitivity analysis of score threshold $\gamma$ in Eq.(2) on the target and open domain of C-Driving.

criteria, resulting in a limited number of features used to update the representative-key and discrepancy features, thereby compromising the representativeness of the discrepancy features in OLDM. By default, we use $\gamma = 0.5$.

We also introduce the score threshold $\gamma$ to ensure the accuracy of pseudo annotations in Eq.(4) of the main paper. In Figure 4, we analyze the segmentation performance under different $\gamma$ settings. Setting $\gamma$ too small (e.g., $\gamma = 0, 0.1, 0.2, 0.3$) introduces noise into the pseudo annotations, leading to erroneous guidance for model training. Conversely, setting $\gamma$ too large (e.g., $\gamma = 0.7, 0.8, 0.9$) disregards the majority of pixels, including some correctly predicted ones. This lack of constraints from the target domain annotations prevents the model from generating accurate annotations for target domain images. By default, we use $\gamma = 0.5$.

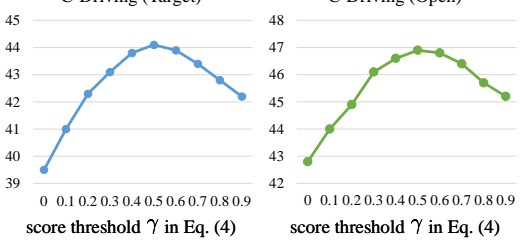

Figure 4: The results of sensitivity analysis of score threshold $\gamma$ in Eq.(4) on the target and open domain of C-Driving.

**Sensitivity Analysis of the selected number of feature sets $K$** Figure 5 illustrates our investigation into various approaches for compensating object features by modifying the number of sets used. When a smaller number of sets is selected for compensation, there is a risk of excluding the sets that correspond to the correct category, resulting in ineffective compensation. Conversely, opting for a larger number of sets introduces error information due to the inclusion of excessive categories. These redundant features offer minimal performance enhancement while increasing testing time. By default, we utilize $K = 9$.

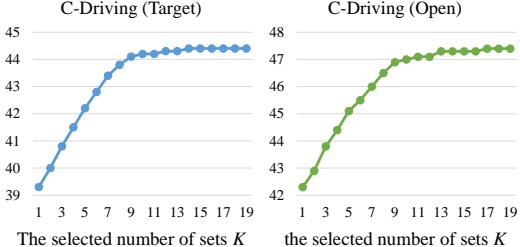

Figure 5: The results of sensitivity analysis of the chosen number of compensation sets $K$ on the target and open domain of C-Driving.

## 2.3  Evaluation of OLDM in Cross-Model Scenarios

Note that OLDM can be incorporated into various segmentation networks. In this context, we assess the performance of OLDM when trained in conjunction with one segmentation network and applied to another network. The results of this evaluation are presented in Table 5. Specifically, we examine the segmentation performance on the C-Driving dataset.

Table 5: The results of the methods exchanging the OLDMs of the model trained with C-Driving and ACDC as target domains, respectively.

|  | Train: GTA5, C-Driving Test: C-Driving | | Train: SYNTHIA, C-Driving Test: C-Driving | |
|---|---|---|---|---|
| w/o OLDM | mIoU(T) | mIoU(O) | mIoU$^{16}$(T) | mIoU(O) |
|  | 36.6 | 39.7 | 28.1 | 36.8 |
| w/o Cross-Model | mIoU(T) | mIoU(O) | mIoU$^{16}$(T) | mIoU(O) |
|  | **44.1** | **46.9** | **35.6** | **48.5** |
| w/ Cross-Model | mIoU(T) | mIoU(O) | mIoU$^{16}$(T) | mIoU(O) |
|  | 44.0 | 46.8 | 35.1 | 48.1 |

We train two of our models using GTA5 and SYNTHIA as source domains, while C-Driving serves as target domain. The performances of these models on the C-Driving target domain are reported in the "w/o Cross-Model" row. Subsequently, we exchange the OLDMs between the two models. For the OLDM of the model trained with SYNTHIA as source domain, we incorporate the missing three categories (terrain, truck, and train) from the OLDM of the model trained with GTA as the source domain. The performances after the exchange are presented in the "w/ Cross-Model" row. Notably, we observe that the model's performance on C-Driving does not significantly decrease following the OLDM exchange. This finding suggests that the model is not highly sensitive to the choice of the source domain.

## 2.4  Analysis on Effect of Compensation

In Figure 6, we resort to t-SNE [10] to visually represent the distributions of three main categories of source features, target features and compensated features in the 2D space. During the training of the network, we obtain sets of source features and target features, respectively. Additionally, by utilizing OLDM, we derive the compensated features corresponding to the target features. The distribution of the source and target features is presented in Figure 6(a), whereas Figure 6(b) illustrates the distribution of the source features and compensated features. It is evident that the distributions of the same category in the target and source features are disparate, whereas the distributions of the compensated and source features exhibit proximity. This observation highlights the efficacy of OLDM in compensating for stylistic differences in target features.

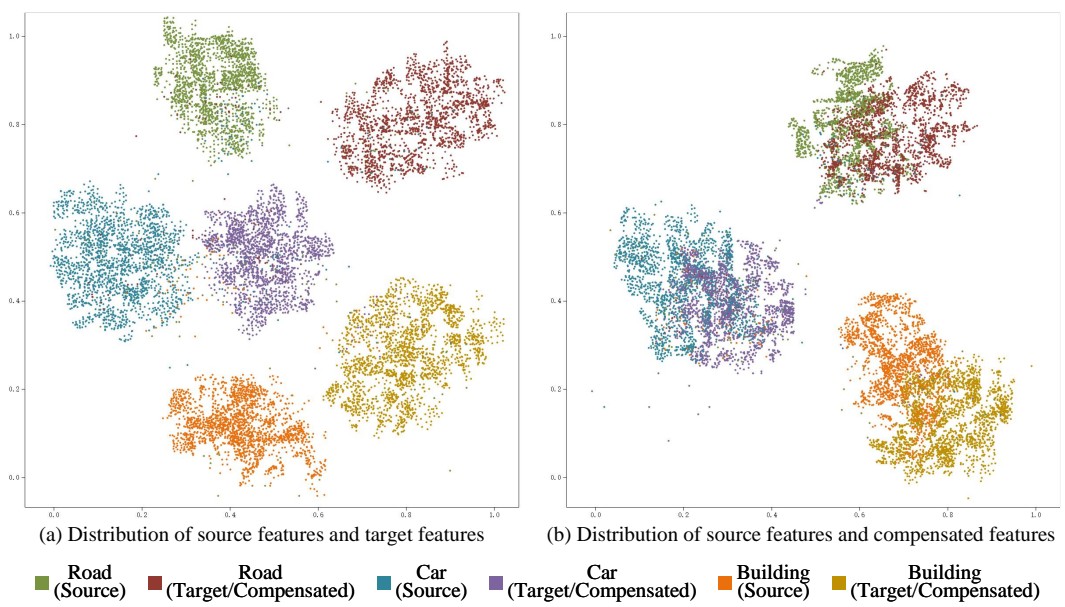

(a) Distribution of source features and target features          (b) Distribution of source features and compensated features

■ Road (Source)   ■ Road (Target/Compensated)   ■ Car (Source)   ■ Car (Target/Compensated)   ■ Building (Source)   ■ Building (Target/Compensated)

Figure 6: Distribution of three categories of source features, target features and compensated features on GTA5(source) and C-Driving(target) datasets. A scatter point represents a feature, which is embedded into the 2D space. The scatter points with the same color represent the features of same categories in source, target/compensated features maps.

## 2.5 More Visual Results

We present additional visual results for C-Driving, ACDC, Cityscapes, KITTI, and WildDash in Figures 8, 9, 10, 11, and 12 at the end of the supplementary material. These visual illustrations showcase the high-quality outputs generated by our segmentation network integrated with OLDM. We also zoom in on some regions from the segmentation results in Figure 7. Our method of object-level style compensation yields satisfactory segmentation results in complicated scenes where the objects are overlapped or occluded.

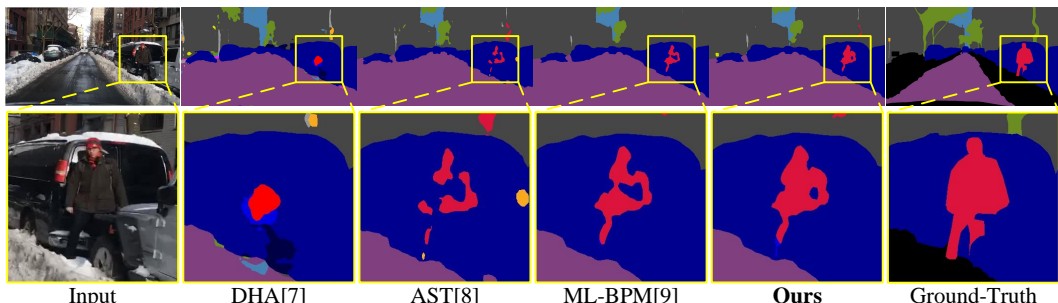

| Input | DHA[7] | AST[8] | ML-BPM[9] | **Ours** | Ground-Truth |

Figure 7: Visualizations of zoom-in segmentation results. We select the overlap pedestrian and cars, whose appearances are too similar, making the segmentation difficult. Our method yields good segmentation results on the overlap.

# 3 Limitation

## 3.1 Failure Cases

In order to offer a comprehensive insight into the limitations of our approach, Figure 13 presents a selection of failure instances. In the first two failure cases, adverse weather conditions have a detrimental impact on the overall image quality, making it challenging to accurately select the appropriate discrepancy features from OLDM. Consequently, this results in subpar outcomes.

In the last two failure cases, certain categories in the input images (e.g., sky and road) exhibit significant deviations from their corresponding category distributions. Despite the inclusion of discrepancy features for each category in OLDM, the distribution disparities of certain categories in the images pose challenges in selecting the corresponding category's discrepancy features from OLDM. Consequently, this difficulty ultimately results in unsatisfactory outcomes.

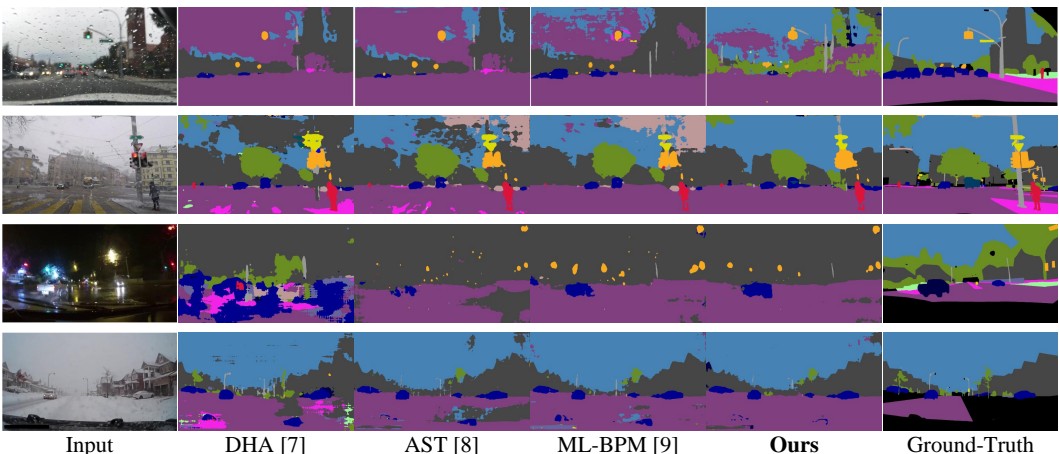

| Input | DHA [7] | AST [8] | ML-BPM [9] | **Ours** | Ground-Truth |

Figure 13: Segmentation results of the failure cases on target and open domains.

## 3.2 Evaluation of OLDM in Cross-Dataset Scenarios

During the experiments, we assess the performance of adaptation and generalization in the target and open domains, respectively. However, we have not examined the adaptation and generalization performance on the same dataset. In this regard, we conduct training using C-Driving and ACDC as target domains and GTA5 as source domain. Subsequently, we perform testing on the respective

target domains of C-Driving and ACDC, as indicated by the entries "w/o Cross-Dataset" and "w/ Cross-Dataset" in Table 6. In the case of "w/ cross-dataset," although we conduct testing on the target domains of ACDC and C-Driving, it can be viewed as an evaluation of generalization performance in an open domain, since C-Driving and ACDC are utilized for network training separately.

In this context, we employ a performance comparison between our approach and the state-of-the-art method ML-BPM in cross-dataset scenarios. As depicted in the Table 6, although both methods experience a decline in performance when subjected to cross-dataset testing, our method manages to achieve superior results compared to the state-of-the-art method. However, it is worth noting that our method exhibits limitations in terms of generalization across datasets, leading to decreased performance in open domain scenarios. Consequently, generalization power of OLDM in cross-dataset scenarios still need to be improved.

Table 6: The results of the methods exchanging the test dataset of two models trained with C-Driving and ACDC as target domains, respectively.

|  | Train: GTA5, C-Driving | | Train: GTA5, ACDC | |
| --- | --- | --- | --- | --- |
|  | Test: C-Driving(T) | | Test: ACDC(T) | |
| w/o Cross-Dataset | ML-BPM[9] | **Ours** | ML-BPM[9] | **Ours** |
|  | 40.2 | **44.1** | 32.1 | **35.7** |
|  | Test: ACDC(T $\rightarrow$ O) | | Test: C-Driving(T $\rightarrow$ O) | |
| w/ Cross-Dataset | ML-BPM[9] | **Ours** | ML-BPM[9] | **Ours** |
|  | 26.4 | 30.2 | 32.7 | 35.2 |

## 4 Code Segment

Our code will be available at: https://github.com/fengtl/OLDM.

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

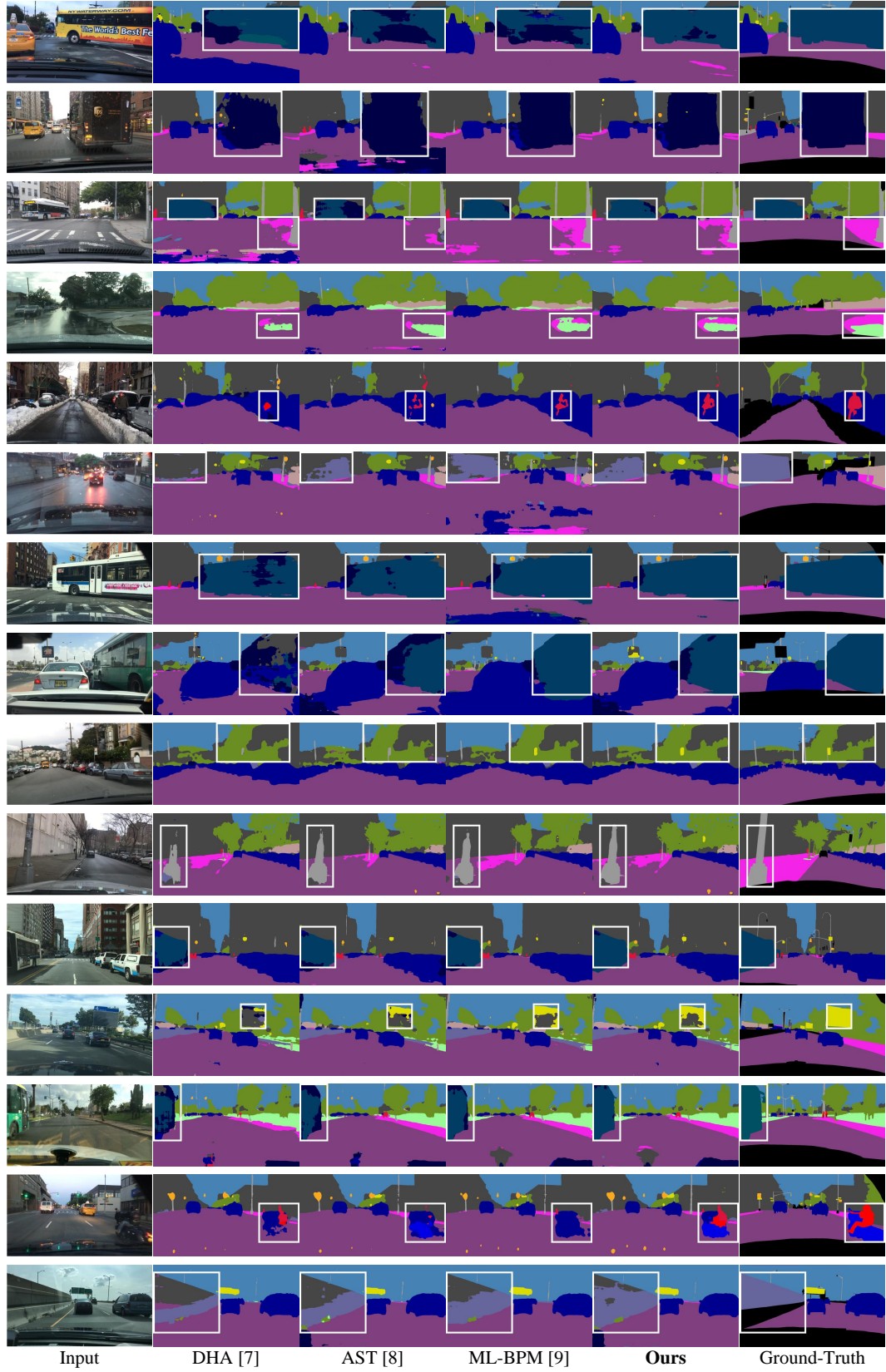

| Input | DHA [7] | AST [8] | ML-BPM [9] | **Ours** | Ground-Truth |

Figure 8: Segmentation results of various methods on the target and open domains of C-Driving.

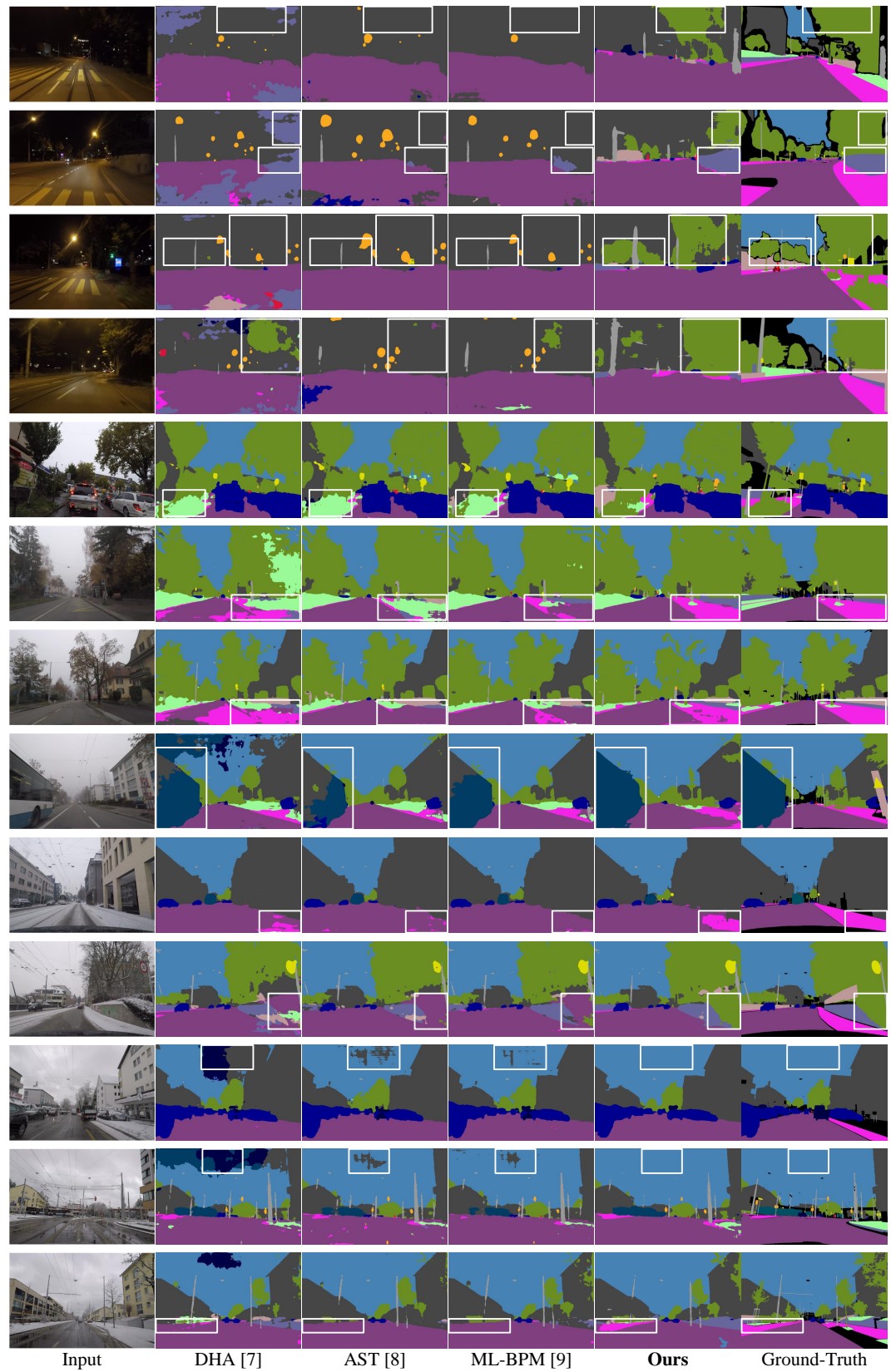

Figure 9: Segmentation results of various methods on the target and open domains of ACDC.

| Input | DHA [7] | AST [8] | ML-BPM [9] | **Ours** | Ground-Truth |

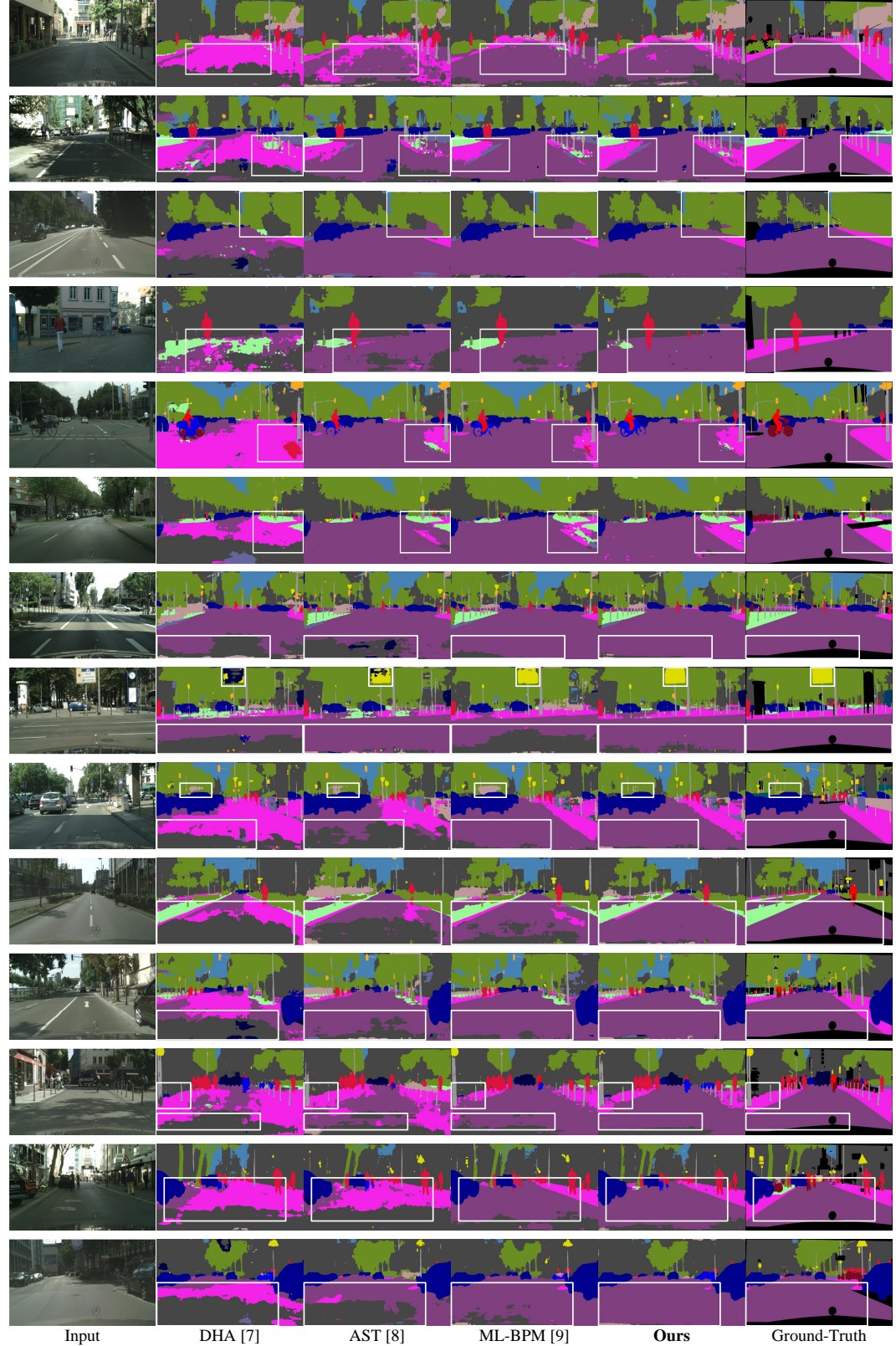

| Input | DHA [7] | AST [8] | ML-BPM [9] | **Ours** | Ground-Truth |

Figure 10: Segmentation results of various methods on the open domain of Cityscapes.

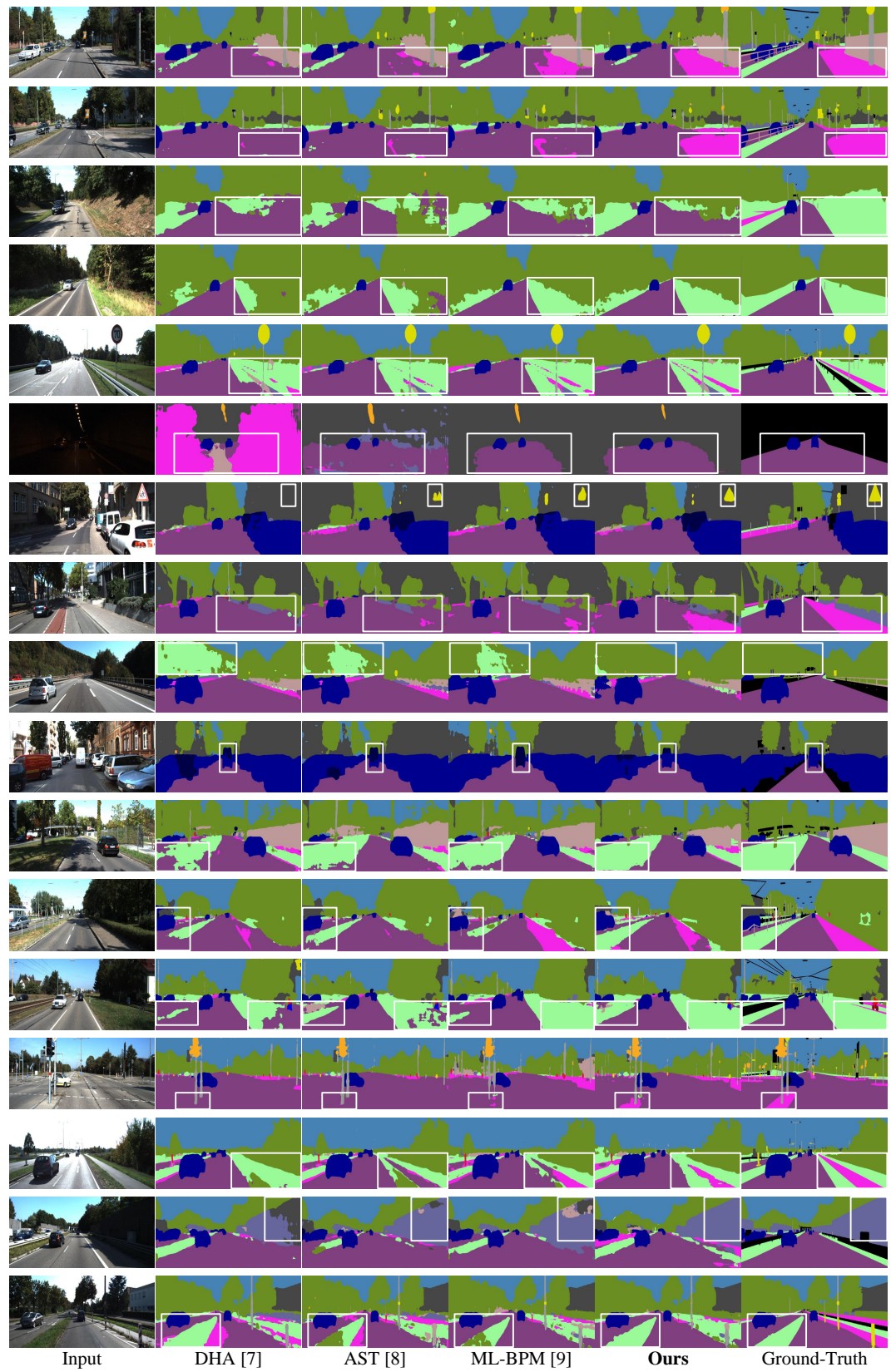

| Input | DHA [7] | AST [8] | ML-BPM [9] | **Ours** | Ground-Truth |

Figure 11: Segmentation results of various methods on the open domain of KITTI.

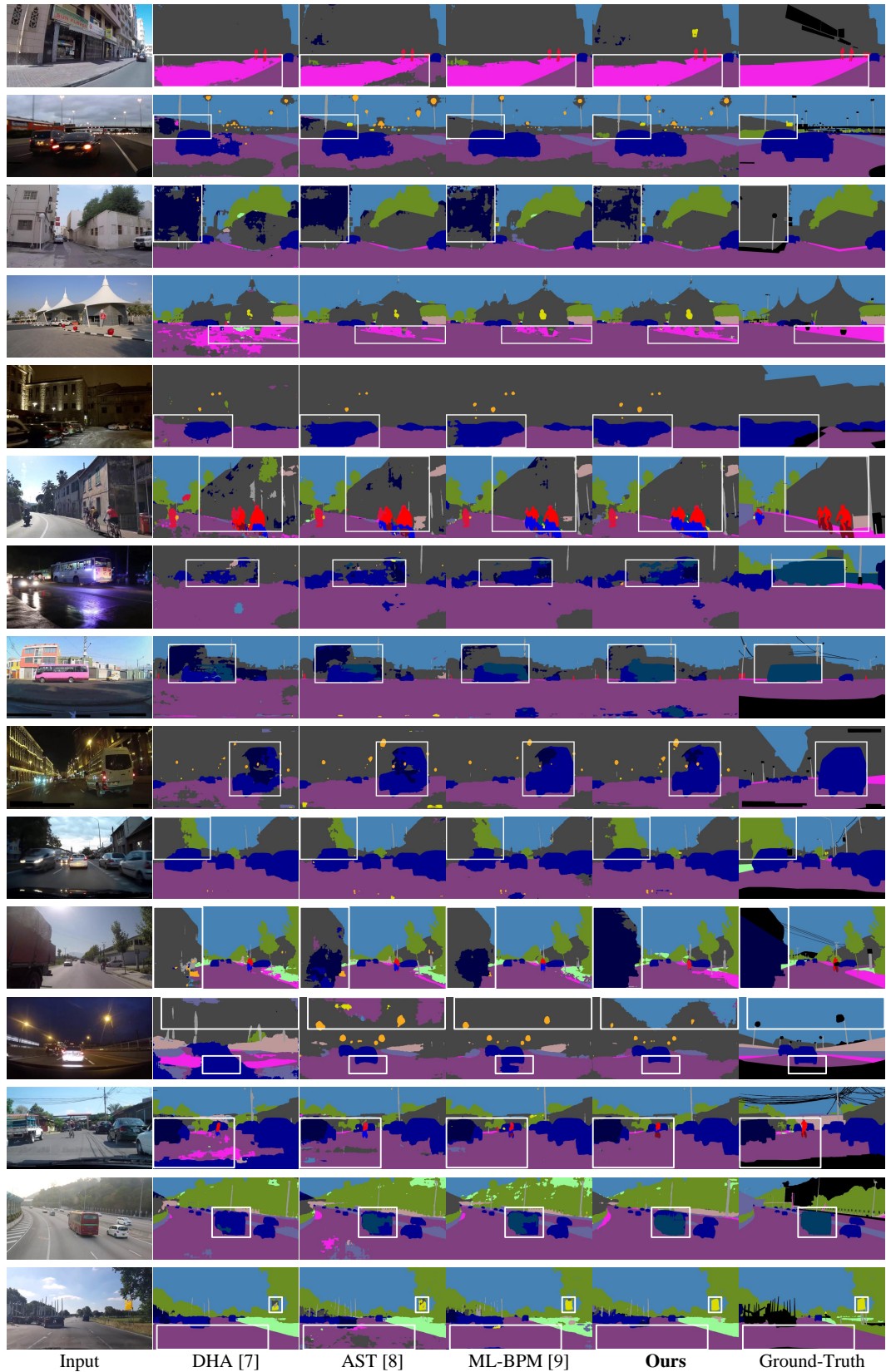

| Input | DHA [7] | AST [8] | ML-BPM [9] | **Ours** | Ground-Truth |

Figure 12: Segmentation results of various methods on the open domain of WildDash.