# OpenReview forum: "Open Compound Domain Adaptation with Object Style Compensation for Semantic Segmentation"
_NeurIPS.cc/2023/Conference — NeurIPS 2023 poster_

### Official Review · Reviewer_CD2C · 2023-06-19

**Soundness:** 3 good
**Presentation:** 4 excellent
**Contribution:** 3 good
**Rating:** 5
**Confidence:** 3

**Summary:**

This paper strives to capture various category-level object style information and then compensate the style information of the object instances from the target to the source domain in the open compound domain adaptation for semantic segmentation. And they evaluate the proposed method on various source and target datasets to verify the robustness and universality.

**Strengths:**

Overall, the writing is well-organized and easy to follow. And the strengths can be summarized as follows:
1. The proposed approach to handling the open compound domain adaptation task from a novel category-level style compensation perspective is interesting and will bring some insights to the community.
2. The components of discrepancy memorization and style compensation are convincing. And the authors conducted comprehensive comparison experiments as well as ablation studies among various benchmarks.

**Weaknesses:**

There are some weaknesses listed as follows:
1. You raise a valid point regarding the terminology "Instance-Key" which may not be appropriate in the context of semantic segmentation tasks where instance-level annotations are not available.
2. You suggest merging the content in Table 7 or moving some sub-tables to the supplementary material to improve the readability and aesthetics of the paper's layout.

**Questions:**

See weaknesses.

---

> ### Author Rebuttal · Authors · 2023-08-08
>
> >### 1. You raise a valid point regarding the terminology "Instance-Key" which may not be appropriate in the context of semantic segmentation tasks where instance-level annotations are not available.
>
> Please see our discussion in the second part of “Response to Common Questions”. The term “instance” of “instance-key feature” does not strictly point to an object instance. An instance-key feature is updated by the same category of object features, which are extracted from different images. The object features from different images naturally represent various object instances. Thus, the instance-key feature is named as such to indicate it is sensitive to the object instances. More specifically, the instance-key feature summarizes the same category’s object instances with similar visual styles. Consequently, an instance-key feature captures one of the representative styles, which widely appear with the object instances of the same category. We plan to revise the “instance-key feature” to the “representative-key feature” to fix the inappropriate name.
>
> >### 2. You suggest merging the content in Table 7 or moving some sub-tables to the supplementary material to improve the readability and aesthetics of the paper's layout.
>
> Thanks for your valuable comment. As the below tables, we plan to merge Table 7(a-b), (c-d), (e-f), respectively, which hopefully improve the paper layout.
>
> **Table(a-b): Train: GTA5(Source)/SYNTHIA(Source), C-Driving(Target). Test: C-Driving(Target).**
> |||GTA5(S)|SYNTHIA(S)|SYNTHIA(S)|
> |----|:----:|:----:|:----:|:----:|
> |Method|Type|mIoU(T)|mIoU$^{16}$(T)|mIoU$^{11}$(T)|
> |Source-only|-|28.3|20.9|28.1|
> |CDAS|OCDA|31.4|25.3|34.0|
> |CSFU|OCDA|34.9|26.1|34.8|
> |DACS|UDA|36.6|28.1|36.5|
> |DHA|OCDA|37.1|29.9|37.6|
> |AST|OCDA|38.8|31.1|38.9|
> |ML-BPM|OCDA|40.2|32.1|40.0|
> |**Ours**|OCDA|**44.1**|**35.6**|**43.7**|
>
> **Table(c-d): Train: GTA5(Source)/SYNTHIA(Source), C-Driving(Target). Test: C-Driving(Open), Cityscapes(Open), KITTI(Open), WildDash(Open).**
> |||||GTA5(S)|||||SYNTHIA(S)|||
> |----|:----:|:----:|:----:|:----:|:----:|:----:|:----:|:----:|:----:|:----:|:----:|
> |Method|Type|CD|CS|KT|WD|Avg|CD|CS|KT|WD|Avg|
> |CSFU|OCDA|38.9|38.6|37.9|29.1|36.1|36.2|34.9|32.4|27.6|32.8|
> |DACS|UDA|39.7|37.0|40.2|30.7|36.9|36.8|37.0|37.4|28.8|35.0|
> |RobustNet|DG|38.1|38.3|40.5|30.8|37.0|37.1|38.3|40.1|29.6|36.3|
> |DHA|OCDA|39.4|38.8|40.1|30.9|37.5|38.9|38.0|40.6|30.0|36.9|
> |AST|OCDA|40.7|40.3|41.9|32.2|38.8|40.5|39.8|41.6|30.7|38.2|
> |ML-BPM|OCDA|42.5|41.7|44.3|34.6|40.8|42.6|41.1|43.4|30.9|39.5|
> |**Ours**|OCDA|**46.9**|**43.6**|**46.5**|**40.1**|**44.3**|**48.5**|**48.0**|**51.3**|**39.6**|**46.9**|
>
> **Table(e-f): Train: GTA5(Source)/SYNTHIA(Source), ACDC(Target). Test: ACDC(Target and Open).**
> |||GTA5(S)|GTA5(S)|SYNTHIA(S)|SYNTHIA(S)|
> |----|:----:|:----:|:----:|:----:|:----:|
> |Method|Type|mIoU(T)|mIoU(O)|mIoU$^{16}$(T)|mIoU$^{16}$(O)|
> |Source-only|-|20.5|27.1|19.8|20.5|
> |CDAS|OCDA|25.3|29.1|25.9|23.3|
> |CSFU|OCDA|27.6|30.5|26.7|24.8|
> |DACS|UDA|29.0|34.8|28.3|27.0|
> |DHA|OCDA|29.5|37.5|29.2|27.3|
> |AST|OCDA|30.7|39.2|30.1|27.9|
> |ML-BPM|OCDA|32.1|41.6|31.9|29.1|
> |**Ours**|OCDA|**35.7**|**44.1**|**34.7**|**36.4**|

---

> > ### Comment · Reviewer_CD2C · 2023-08-17
> >
> > Thanks to the author for solving most of my concerns. I choose to keep my score as borderline accept.

---

> > > ### Author Response · Authors · 2023-08-18
> > > **Thanks for Your Review**
> > >
> > > Dear Reviewer CD2C,
> > >
> > > Thank you again for your review. We are pleased to see that the questions raised by you are solved.
> > >
> > > Best,
> > >
> > > Authors of Paper ID 10143

---

### Official Review · Reviewer_okqW · 2023-07-05

**Soundness:** 2 fair
**Presentation:** 2 fair
**Contribution:** 2 fair
**Rating:** 3
**Confidence:** 4

**Summary:**

This paper deals with the open compound domain adaptive semantic segmentation problem. The motivation is to compensate the object style gap across domains to obtain more accurate pseudo labels for self-training. The main framework consists of two parts:  discrepancy memory and style compensation. For discrepancy memory, the aim is to memorize the category-level and instance-level feature discrepancy between target and source features. For style compensation, the aim is to compensate the style variation from source to target to obtain high-quality pseudo labels to facilitate the subsequent self-training procedure. Finally, two cross-entropy loss terms with ground-truth source annotations and target pseudo labels respectively are minimized to update the segmentation network. Experiment results show that the proposed method not only improves the segmentation performance on the target domain, but also yields better segmentation results on the open domain, compared to previous methods.

**Strengths:**

- The idea is simple and straightforward.
- The paper is generally easy to follow and understand.
- Ablations of various alternative solutions are conducted.

**Weaknesses:**

- Some details are not clearly stated. For example, what is the intermediate segmentation head? From the start to the end of the paper, I don't find a formal definition of the open compound domain adaptive semantic segmentation task. A clear definition of the task may help the reader to better understand the proposed method.

- The proposed method involves many hyper-parameters, e..g., the $\lambda$ in Eq. (1); the $\gamma$ in Eq. (2) and Eq. (4); the $K$ in Eq. (3), etc. As we know, for a domain adaptation problem, it is hard to perform the hyper-parameter selection. I just doubt how those hyper-parameters affect the final performance. The authors should provide empirical studies and analysis.

-  For the Instance-Key and Discrepancy Features, they are instance-level (see Eq. (2)). I just doubt if such method can be scalable to large-scale datasets, as the $m$ may be super large (there are large numbers of pixels within one specific category).

- For the ablation part, I suggest the authors use equations along with text to better describe each alternative solutions. Only with text descriptions, it is not easy for the reader to accurately capture the meaning that the authors want to express.

- Some necessary ablations are missing. For example, in Eq. (7), how about moving the first term and just using the last term? In Eq. (2) and Eq. (3), how about not using the weighting mechanism, instead simply using the average? For Eq. (1), how about simply averaging features without $\lambda$?

- For Eq. (1), it actually may change the scale of feature values as time goes on. I just doubt why this could work. Maybe moving average is more suitable.

- From the experiment results, I see the mIoU for the open domain is generally higher than the target domain, though open domain images are not utilized during training. Do the authors have any explanations?

- From a high-level, I doubt if this way can really help the segmentation on the open domain, as the compensation is designed for the target, i.e., based on the distribution or statistics of target images or features. How can such compensation generalize to an unseen domain?

- The paper should be polished further to make the descriptions more professional and accurate.

**Questions:**

See the weakness part.

**Limitations:**

I don't see serious societal issues.

---

> ### Author Rebuttal · Authors · 2023-08-09
>
> >### 1. Some details are not stated, like the intermediate segmentation head and the definition of OCDA for semantic segmentation.
>
> Figure 2 of the rebuttal file illustrates the intermediate head as the convolutional layers. This head takes input as the target feature for regressing the category score map.
>
> The existing works (i.e., ML-BPM [ECCV 2022], CSFU [CVPR2021], and DHA [NeurlPS2020]) have defined OCDA for semantic segmentation. In training, the segmentation network learns from the source domain, where the images have pixel-level annotations. The network also learns from the target images with different styles from the source images. The target images are given without annotations. Thus their pseudo annotations should be regressed for the network training. In testing, the segmentation network is evaluated on the open domain images with styles unseen in training. We will add the definition to the paper.
>
> >### 2. The method involves many hyper-parameters, which are hard to select. How do they affect the performance?
>
> In Section 2.2 of the supplementary file, we have studied the hyper-parameters (i.e., the memory capacity $M$ in Eq. (2), the ratio $\lambda$ in Eq. (1), the score threshold $\gamma$ in Eqs. (2) and (4), the feature set number $K$ in Eq. (3)). The performances in Figures 1-5 of the supplementary file guide us to select the hyper-parameters, which leads to the best results on the C-Driving dataset are used for evaluating our method on the Cityscapes, KITTI, WildDash, and ACDC datasets. Though it is hard to select hyper-parameters, our approach with robustness uses the same set of hyper-parameters to outperform other methods on various datasets.
>
> >### 3. For the Instance-Key and Discrepancy Features, they are instance-level (see Eq. (2)). Can such a method be scalable to large-scale datasets, as the $m$ may be super large?
>
> “$m=1,…,M$” is the index of the instance-key feature, which captures one of the representative styles appearing with the same category's instances. “$M=50$” already leads to satisfactory performances on various datasets (see the second part of “Response to Common Questions”).
>
> >### 4. The authors should use equations with text to describe each ablation alternative.
>
> Thanks. We plan to add the equations with Tables 2-6 as below.
>
> w/o OLDM: w/o Eqs. (1)(2)(3)
>
> global(T2), 100% update(T3), multi-sets, category(T4), instance similarity(T5), final(T6): w/ all Eqs. in paper
>
> T2
>
> mean instances$$A_l = \frac{1}{M} \sum_m N_{l,m}$$
> local$$A_l = \frac{1}{bs} \sum_{bs} F_s(x,y)$$
> T3
>
> mean discrepancy$$D_{l,m} \leftarrow \frac{N\cdot D_{l,m} + (A_l - F_t(x,y))}{N+1},\widetilde{F_t}(x,y) = F_t(x,y) + \sum_k \sum_mD_{k,m}$$
> discrepancy similarity$$w_{l,m} = \frac{D_{l,m} \cdot F_t(x,y)}{\sqrt{C}}$$
> top-1 update$$m = argmax \\{w_{l,m}\\}, w_{l,m}=\frac{N_{l,m} \cdot F_t(x,y)}{\sqrt{C}}$$
> top-50% update$$m\in max_{\frac{M}{2}}\\{w_{l,m}\\}, w_{l,m}~~as~above$$
> T4
>
> key discrepancy$$D_{l,m} = A_l - N_{l,m}$$
> merged sets$$A \leftarrow A + \lambda F_s(x,y),N_m \leftarrow N_m + w_mF_t(x,y),D_m \leftarrow D_m + w_m(A-F_t(x,y)),w_m = \frac{N_{m} \cdot F_t(x,y)}{\sqrt{C}}$$
> multi-sets, k-means$$C_i=\\{D_m | d(D_m, c_k)\ge d(D_m,c_i)~~for~k\neq i\\}, c_i=\frac{1}{|C_i|}\sum_{D_m\in C_i}D_m$$
> T5
>
> mean discrepancy$$\widetilde{F_t}(x,y) = F_t(x,y) + \sum_mD_{k,m}$$
> T6
>
> w/o pseudo: w/o Eq. (4)
>
> intermediate: replace $\widetilde{{\bf R}}$ with ${\bf R}$ in Eq. (4)
>
> >### 5. Ablations are missing. Eq. (7) w/o the first term? Eqs. (2-3) w/o weighting but the average? Eq. (1) w/o $\lambda$ but the averaging?
>
> By removing the first term of Eq. (7), we turn off the intermediate segmentation head in training, missing a chance to update the shallow object feature map $F_t$. It degrades the result on the C-Driving dataset from 44.1 to 38.4 mIoU.
>
> In Eqs. (2-3), the weights measure the similarities between the object and instance-key features. They re-weight the discrepancy features to compensate for the object features of the target/open domain to the source domain. By replacing the weighting with the moving average, Eq. (3) fails to select the appropriate discrepancy features, degrading the result from 44.1 to 40.7 mIoU. Please also see the first part of “Response to Common Questions”.
>
> In Eq. (1), we make little difference to the result by replacing $\lambda$ with the moving average (44.1 v.s. 43.9 mIoU). Empirically, $\lambda=0.001$ prevents the outlier object features of the source domain from primarily impacting the category-key features. It lets the category-key features converge faster than the moving average (250K v.s. 350K iterations).
>
> >### 6. Eq. (1) may change the scale of features as time goes on. Why this work? The moving average is more suitable.
>
> Please see the last paragraph of our response to your fifth concern.
>
> >### 7. mIoU for open domain is generally higher than the target domain, though the open domain is not for training. Any explanation?
>
> With the source domain of GTA5, better performances on the open domain of ACDC than the target domain have been found in the many methods (e.g., CDAS [CVPR 2020], AST [AAAI 2022], and ML-BPM [ECCV 2022] in Table 7(e)). Though this comparison between distinct domains with different images is unfair and out of the evaluation scope, this is partially because the open domain is near the source domain, where the features are easily transformed to. With the source domain of SYNTHIA far away from the open domains, this phenomenon is generally not found. Even so, our method excellently transforms the open object features to the source domain, yielding better performances than the target domain.
>
> >### 8. If this way can help the segmentation in the open domain, as the compensation is designed based on the distribution or statistics of target images or features? How can it generalize to an unseen domain?
>
> Please see the second part of “Response to Common Questions”.
>
> >### 9. The paper should be polished.
>
> Thanks. We will refine our paper.

---

> > ### Comment · Reviewer_okqW · 2023-08-21
> >
> > Thanks for the authors' effort for providing detailed response. After reading the response, I still feel confused about the reason why the compensation designed based on the distribution or statistics of target images or features can generalize to unseen domains.

---

> > > ### Author Response · Authors · 2023-08-21
> > > **Thanks for your response**
> > >
> > > Thanks for your response! Please note that we provide the reason why the memory for storing the discrepancy features can help the generalization to unseen domain. Below, we provide a summary for you to better understand it.
> > >
> > > The compensation designed based on the distribution or statistics of target images or features mentioned by you means using the memory to store the discrepancy features for compensating the target images to the source domain. Actually, the discrepancy features represent the difference between the source and target images, rather than representing the target images only. Thus, the discrepancy features build the **clear association** between the target and source images. The advantage of the discrepancy features for generalization to the unseen domain is two-fold:
> > >
> > > - **A Better Representation of Category-level Style Difference across Target and Source Domains**. With a **clear association** between the source and target images, the discrepancy features can represent the difference between the object styles of the same category but across the source and target domains. This can be easily done because images have pixel-wise annotations of semantic categories. It means the discrepancy features are specified to the corresponding categories, thus representing the category-level style difference across source and target domains.
> > > - **A More Generalized Representation of Category-level Style Difference across Unseen and Source Domains**. Given an object also in the same category to target/source domain but in an unseen domain, we can use its feature to compute the similarities with the object features of target images. Note that these similarities also have a **clear association** with the discrepancy features, because they are computed based on the target and unseen object features. They can re-weight the associated discrepancy features, forming the **new discrepancy feature** to compensate the unseen style, which is transformed to the source domain. **Please note that the new discrepancy feature is the core for the generalization to the unseen domain specified to the category**.
> > >
> > >
> > > **Simply put, the success of our method stems from using the discrepancy features attending to the semantic categories, which build the clear association between the style difference across target-source domains, and the similarity between unseen-target domains, finally forming the new and generalized discrepancy feature to transform a category’s style of the unseen domain to the source domain**. Please check the visualized process of our method in Figure 1(b) of our rebuttal file (https://openreview.net/attachment?id=LOvUNcunkJ&name=pdf). In comparison, the existing methods trivially depend on the parameters of the deep network to transform the image features across different domains, failing to achieve the above success. This is because the network contains many layers of parameters. How can the parameters be associated to each semantic category, or to unknown number of target and unseen domains in different layers? How can we re-weight and re-combine these parameters without an explicit association between the parameters and the target domains? Without properly addressing these problems, the existing methods cannot achieve the **clear association** between the **category-level style difference** across target-source domains, and the similarity between unseen-target domains, thus showing worse generalization to the unseen domain than our method.

---

> ### Author Response · Authors · 2023-08-19
> **Sincerely Request Your New Comment**
>
> Dear Reviewer okqW,
>
> We thank you again for your valuable comments, which significantly help us to polish our paper. We are looking to discussing with you the questions that are addressed unsatisfactorily.
>
> Best,
>
> Authors of Paper ID 10143

---

> > ### Author Response · Authors · 2023-08-21
> > **Sincerely Request Your New Comment Again**
> >
> > Dear Reviewer okqW,
> >
> > Again, please allow us to extend our sincere thanks to you, for your time and efforts of reviewing our paper. As the deadline for the authors' response is approaching, we sincerely request your comment on our primary response. This will definitely give us a valuable chance to address the questions unsolved.
> >
> > Best,
> >
> > Authors of Paper ID 10143

---

> ### Comment · Area_Chair_Nqi8 · 2023-08-21
>
> Dear Reviewer okqW,
>
> Could you give a quick feedback to authors' rebuttal? Thank you!
>
> Best regards,
> AC

---

### Official Review · Reviewer_XmuL · 2023-07-06

**Soundness:** 3 good
**Presentation:** 3 good
**Contribution:** 3 good
**Rating:** 6
**Confidence:** 5

**Summary:**

The paper introduces Object Style Compensation which involves constructing an Object-Level Discrepancy Memory consisting of multiple sets of discrepancy features to minimize the style differences between the source and target domains and ensure the styles of different object categories or instances within the scene to be adapted well. In particular, the paper learns these discrepancy features from images of both domains and stores them in memory. By leveraging this memory, appropriate discrepancy features are selected to compensate for the style information of object instances across various categories, aligning their styles to a unified style of the source domain. This method enables more accurate computation of pseudo annotations for target domain images and achieves state-of-the-art results on different datasets.

**Strengths:**

- Overall the paper is well organized and easy to read.

- The idea of using the object-level discrepancy memory for domain adaptation is interesting.

- Achieved better semantic segmentation performance than SOTA methods.


**Weaknesses:**

- What's the value of the parameter $\gamma$ set in Eqn (2) and (4), and how to determine its value? I would like to suggest the authors to run a group of experiments with different values of $\gamma$ to see how the value impacts the final performance.

- The implementation detail is missing in the main paper. I would like to suggest the authors move this part from the supplementary to the main paper so that the reader can well understand and even reproduce the results if necessary.

- The current paper still misses one important baseline, i.e., using the category-level discrepancy rather than the instance-level.

- For the visualization part, the visualization in the current paper is still very simple. I am curious whether the current method can handle the complicated scenes (e.g., with overlap and occlusion between objects with same or different categories).


- The authors didn’t provide the failure cases and the corresponding analysis in the current paper.


**Questions:**

1. What’s the runtime for the current method?

2. What are the failure cases?


**Limitations:**

The author didn’t discuss the limitations of the current proposed method.

---

> ### Author Rebuttal · Authors · 2023-08-08
>
> >### 1. What's the value of the parameter $\gamma$ set in Eqn (2) and (4), and how to determine its value? I would like to suggest the authors to run a group of experiments with different values of $\gamma$ to see how the value impacts the final performance.
>
> Thanks. We have provided this analysis in Section 2.2 “Sensitivity Analysis of score threshold \gamma” of the supplementary material. Figures 3 and 4 of the supplemental material report the performances with different values of \gamma on the C-Driving dataset. The $\gamma$, which leads to the best performance on the C-Driving dataset, is finally used for evaluating our method on different datasets.
>
> >### 2. The implementation detail is missing in the main paper. I would like to suggest the authors move this part from the supplementary to the main paper so that the reader can well understand and even reproduce the results if necessary.
>
> Thanks. We will move the necessary details to the main paper according to your suggestion.
>
> >### 3. The current paper still misses one important baseline, i.e., using the category-level discrepancy rather than the instance-level.
>
> Thanks. The table below compares the performances achieved by the category- and level-level discrepancy features. Compared to the baseline without style compensation (“w/o OLDM”), the category-level discrepancy features (“category-level discrepancy”) help to transform the target and open domains’ object features to the source domain, finally yielding better performances. However, the category-level discrepancy features insufficiently account for the style difference between the object instances of the target/open and the source domains. Thus, their performances lag behind our method with the instance-level discrepancy features.
>
> | method | mIoU(Target) | mIoU(Open) |
> | :-: | :-: | :-: |
> | w/o OLDM | 36.6 | 39.7 |
> | category-level discrepancy | 38.1 | 40.2 |
> | Ours | 44.1 | 46.9 |
> |
>
> >### 4. For the visualization part, the visualization in the current paper is still very simple. I am curious whether the current method can handle the complicated scenes (e.g., with overlap and occlusion between objects with same or different categories).
>
> Thanks. In Figure 3 of the rebuttal file, we zoom in on some regions from the segmentation results. Our method of object-level style compensation yields satisfactory segmentation results in complicated scenes where the objects are overlapped or occluded. We will add these results to the main paper and supplementary file.
>
> >### 5. The authors didn’t provide the failure cases and the corresponding analysis in the current paper.
>
> We respectfully clarify that the discussion on the failure cases has been provided in Section 3.1 of the supplementary file. In this discussion, images with adverse weather conditions lead to unsatisfactory results. The failure cases also appear when the images of the open domains exhibit significant deviations from the source domain. These factors make it challenging to select the appropriate discrepancy features from OLDM. Though our method outperforms other methods, we will investigate how to reduce failure cases in challenging scenarios.
>
> >### 6. What’s the runtime for the current method?
>
> In Figure 1(c) of the supplementary file, we study the testing time of our method by changing the capacity of OLDM. With the default capacity (M=50), we need 0.380 seconds on average to process a 1280x720 image. We also compare the testing time of our method with other methods in the table below. Our method achieves better performance at the cost of reasonable testing times (second per each 1280x720 image).
>
> | method | source-only | CDAS | CSFU | DACS | RobustNet | DHA | AST | ML-BPM | Ours |
> | :-: | :-: | :-: | :-: | :-: | :-: | :-: | :-: | :-: | :-: |
> | running time |0.345 |0.387 |0.383 |0.364 |0.378 |0.397 |0.411 |0.402 |0.380 |
>
> >### 7. The author didn’t discuss the limitations of the current proposed method.
>
> Please see Section 3 of the supplementary file, where we have analyzed the limitations of our approach. This analysis comprises of two aspects: "Failure Cases" and "Evaluation of OLDM in Cross-Dataset Scenarios".

---

> ### Author Response · Authors · 2023-08-19
> **Sincerely Request Your New Comment**
>
> Dear Reviewer XmuL,
>
> We thank you again for your valuable comments, which significantly help us to polish our paper. We are looking to discussing with you the questions that are addressed unsatisfactorily.
>
> Best,
>
> Authors of Paper ID 10143

---

> > ### Author Response · Authors · 2023-08-21
> > **Sincerely Request Your New Comment Again**
> >
> > Dear Reviewer XmuL,
> >
> > Again, please allow us to extend our sincere thanks to you, for your time and efforts of reviewing our paper. As the deadline for the authors' response is approaching, we sincerely request your comment on our primary response. This will definitely give us a valuable chance to address the questions unsolved.
> >
> > Best,
> >
> > Authors of Paper ID 10143

---

### Official Review · Reviewer_uGsB · 2023-07-07

**Soundness:** 3 good
**Presentation:** 2 fair
**Contribution:** 3 good
**Rating:** 5
**Confidence:** 5

**Summary:**

The authors propose a novel target-to-source feature style transfer approach for open compound domain adaptation. Inspired by the observation of the existence of object style discrepancy when converting a target image to a source style, they design Object-Level Discrepancy Memory (OLDM). This module saves the category-key features of the source domain, instance-key features of the target domain, and discrepancy features between source and target domains. When performing style transfer from target features to source features, they compensate for the object inconsistency using discrepancy features specific to each object. They evaluate their method on various driving datasets in semantic segmentation.

**Strengths:**

Their motivation is good and intuitive. I agree that there will be a certain degree of appropriate object style for each domain.

It appears that well-designed modules are in place based on motivation.

**Weaknesses:**

In the related work section, there is a simple listing of various papers, and the flow of each research is not well organized. Furthermore, there is little mention of papers published after 2021, which seems to indicate a lack of investigation into recent papers.

Based on the proposed method alone, it seems that it is not specifically designed for open compound domain adaptation but rather for multi-target domain adaptation. The difference between the two tasks lies in whether there is a consideration for how to generalize to unseen domains, which does not appear to be addressed in this paper. The proposed target-to-source feature style transfer method is suitable for multi-target domain adaptation as it aims to obtain more accurate pseudo labels for each target domain. [1] presents a concept of transferring source features into various styles to achieve domain generalization. By incorporating additional well-designed techniques such as source feature style randomization, it can be considered a suitable method for open compound domain adaptation.

This research focuses on semantic segmentation, but the method requires instance masks. However, the paper does not explain how to obtain instance masks.

Minor issues.
- The content in Figure 2 seems to be sufficiently understandable from Figure 1-(c). It would be better to combine Figures 2 and 3 appropriately and allocate more space to the related work section.
- There are three duplicated reference numbers of '23' on line 65."


[1] WildNet: Learning Domain Generalized Semantic Segmentation from the Wild (CVPR'22)

**Questions:**

As I mentioned in the weaknesses section, this work seems to be most related to multi-target domain adaptation. Could the authors explain this research from the perspective of domain generalization?

While proposing source-to-target feature style transfer in addition could potentially offer performance advantages, what is the reason behind only proposing the target-to-source approach?

This method seems to require instance masks, but how do we obtain instance masks from a dataset that only has semantic masks?

**Limitations:**

They address the limitations of their work in supplementary materials.

---

> ### Author Rebuttal · Authors · 2023-08-08
>
> >### 1. In the related work section, there is a simple listing of various papers, and the flow of each research is not well organized. Furthermore, there is little mention of papers published after 2021, which seems to indicate a lack of investigation into recent papers.
>
> Thanks. We have added 14 publications since 2021 to the related work. We plan to reorganize the related work as below.
>
> In the flow of "domain adaptation for semantic segmentation", the unsupervised domain adaptation (UDA) and multi-target domain adaptation train networks through the style regression (SMPPM [AAAI2022]), domain adversarial (DWL [CVPR2021], CCL [CVPR2021]), and self-training (FixBi [CVPR2021], MTKT [ICCV2021]). In contrast, the domain generalization (DG) methods involve learning domain invariant representation (T3A [NeurIPS2021], COMEN [CVPR2022], PCL [CVPR2022]) and data augmentation (L2D[ICCV2021], MBDG[NeurIPS2021], StyleNeophile[CVPR2022]). Regarding various methodologies for open compound domain adaptation (OCDA), they conduct the style regression based on scene-level (DHA [NeurIPS2020], CSFU [CVPR2021], AST [AAAI2022], ML-BPM [ECCV2022]) and category-level (CDAS [CVPR2020]) attributes.
>
> In the flow of "deep network with memorization for visual understanding", the memorization technique is demonstrated in the context of image (CICM [NeurIPS2022], ATDOC [CVPR2021], PinMemory [CVPR2022]) and video (HF2-VAD [ICCV2021], XMem [ECCV2022]) tasks. Additionally, based on the granularity at which objects are stored in memory, these methods can be categorized at either the scene level (MOSS [CVPR2021], CICM [NeurIPS2022], XMem [ECCV2022]) or the category level (ATDOC [CVPR2021], PinMemory[CVPR2022]).
>
> >### 2. The proposed method is not specifically designed for open compound domain adaptation but rather for multi-target domain adaptation. How to generalize to unseen domains is not addressed.
>
> Our method of style compensation is specifically designed to transform the object features of the target and open domains to the source domain. In the first part of “Response to Common Questions”, we explain why our method can be generalized better to address the image segmentation on the unseen domains than the popular methods.
>
> >### 3. WildNet presents a concept of transferring source features into various styles to achieve domain generalization. By incorporating additional well-designed techniques such as source feature style randomization, it can be considered a suitable method for open compound domain adaptation.
>
> Thanks for pointing out this relevant work, which will be added to our paper for discussion. WildNet borrows the object features of the wild domain, which is usually provided by large-scale datasets (e.g., ImageNet), to enrich the object features of the source domain. WildNet extends the source domain, where the object features can cover the unseen domains. Though this extension is effective, it is infeasible to cover all unseen domains exhaustively.
>
> In contrast to WildNet, we follow another research direction of addressing OCDA. We store the discrepancy features between the source and target domains for style compensation, which can be reasonably generalized to transform the object features of the open domain to the source domain. Note that these works along different directions can work together to achieve better performances. Intuitively, the object features extended by WildNet belong to the target domains newly created. We add these extended features to the object features in the target domains in the experimental datasets. Next, we trivially compute the discrepancy features based on the object features of the source and target domains, where the discrepancy features are used for compensating the features of the open domain. In the table below, we compare WildNet, style compensation, and their combination, where the last method achieves the best performances on different datasets.
>
>
> | method | type | C-Driving(Open) | ACDC(Open) | Cityscapes(Open) | KITTI(Open) | WildDash(Open) |
> | - | :-: | :-: | :-: | :-: | :-: | :-: |
> | WildNet | DG | 42.3 | 40.8 | 40.7 | 41.5 | 34.2 |
> | Ours | OCDA | 46.9 | 44.1 | 43.6 | 46.5 | 40.1 |
> | Ours+WildNet | OCDA | 48.3 | 45.8 | 44.1 | 47.4 | 42.7 |
>
> >### 4. This method seems to require instance masks, but how do we obtain instance masks from a dataset that only has semantic masks?
>
> We clarify that instance masks are unnecessary in our method. Please see the second part of “Response to Common Questions”. We plan to revise the “instance-key feature” to the “representative-key feature” to fix the inappropriate name.
>
> >### 5. The content in Figure 2 seems to be sufficiently understandable from Figure 1(c). It would be better to combine Figures 2 and 3 appropriately and allocate more space to the related work section.
>
> Thanks. Figure 1(c) shows the importance of considering the style difference between different object categories and instances. It provides a high-level motivation for proposing the style compensation of different categories and instances. But it contains little technical information. Thus we need Figures 2 and 3 to provide technical details, We agree that Figures 2 and 3, whose redundant elements can be trimmed to better understand the technical pipeline. Please see the revised and merged figures in Figure 2 of the rebuttal file.
>
> >### 6. There are three duplicated reference numbers of '23' on line 65."
>
> Thanks. We have fixed the duplication.
>
> >### 7. What is the reason behind only proposing the target-to-source approach?
>
> With the target-to-source compensation, we transform the object features of the target domains to have a similar style to the source domain. Consequently, the segmentation network only needs to learn from the visual styles of the source domain, producing reliable pseudo labels for the images of the target domains with more variant styles. This advantage cannot be achieved by conducting source-to-target compensation.

---

> > ### Comment · Reviewer_uGsB · 2023-08-21
> >
> > The authors' thorough responses are greatly appreciated. As most of my concerns are resolved, I would raise my score to "Borderline accept".

---

> > > ### Author Response · Authors · 2023-08-21
> > > **Thanks for Your Review**
> > >
> > > Dear Reviewer uGsB,
> > >
> > > Thank you again for your valuable comments. We are pleased to see that the questions raised by you are solved and the score is promoted by you to "Borderline accept". **Please kindly allow us to remind that the score is changed via the syetem entrance of "Rating", which will affect the final decision.**
> > >
> > > Best,
> > >
> > > Authors of Paper ID 10143

---

> ### Author Response · Authors · 2023-08-19
> **Sincerely Request Your New Comment**
>
> Dear Reviewer uGsB,
>
> We thank you again for your valuable comments, which significantly help us to polish our paper. We are looking to discussing with you the questions that are addressed unsatisfactorily.
>
> Best,
>
> Authors of Paper ID 10143

---

### Official Review · Reviewer_LZYW · 2023-07-13

**Soundness:** 3 good
**Presentation:** 2 fair
**Contribution:** 3 good
**Rating:** 6
**Confidence:** 3

**Summary:**

This paper introduces a new method for OCDA called Object Style Compensation, which focuses on adapting the style changes of different categories or instances of objects rather than just the overall scene style. This stored information is used to select the appropriate discrepancy features for compensating the style information of object instances. Then, object features are compensated using selected discrepancy feature. The authors construct the Object-Level Discrepancy Memory by storing multiple sets of discrepancy features. The proposed method achieves SOTA on OCDA benchmarks.

**Strengths:**

- Using the style of objects is both ideal and intriguing.
- Experiments show the effectiveness of the proposed method.

**Weaknesses:**

- How exactly are the objects obtained? It's understandable that categories can be distinguished by pseudo labels, but it's unclear how different instances within the same class are differentiated.
- Why is storing style differences important? Are there any advantages to storing these discrepancies rather than just performing style regression?

**Questions:**

- The explanation on L110 indicates that 'l' represents the category, but there is no explanation about what 'm' signifies. Does 'm' mean the number of instances?
- The quality of the pseudo labels seems likely to influence Discrepancy Memorization. Is there a guarantee of a certain level of quality even in the initial stages of learning?

**Limitations:**

There are no limitations to potential negative societal impact.

---

> ### Author Rebuttal · Authors · 2023-08-08
>
> >### 1. It's unclear how different instances within the same class are differentiated.
>
> Thank you for pointing out this confusion between the instance-key feature and the object instance. In this paper, we do not need to differentiate the instances as those in the instance segmentation task. Please see the second part of “Response to Common Questions”. We plan to revise the “instance-key feature” to the “representative-key feature” to fix the inappropriate name.
>
> >### 2. Are there any advantages to storing these discrepancies rather than just performing style regression?
>
> We respectfully point out that the style regression (e.g., ML-BPM [ECCV 2022], AST [AAAI 2022], CSFU [CVPR2021], DHA [NeurlPS2020], and CDAS [CVPR2020]) unsatisfactorily addresses Open Compound Domain Adaptation (OCDA), where the styles of different object categories and instances should be appropriately transformed. It motivates us to store the features of the style discrepancies, providing a more explicit and flexible way of changing the styles at the category and instance levels. Please see our discussion in the first part of “Response to Common Questions”.
>
> >### 3. There is no explanation about what “$m$” signifies.
>
> We denote “$m=1,…,M$” as the index of the representative-key feature, where “$M$” is the total number of the representative styles of the instances in the same object category. “$M$” serves as a hyper-parameter. In Section 2.2 “Sensitivity Analysis of Memory Capacity” of the supplementary file, we have experimented with changing “M” and examining the effect on the segmentation performance. We will add the explanation to the main paper.
>
> >### 4. Is there a guarantee of a certain level of quality even in the initial stages of learning?
>
> To reduce the negative impact of low-quality pseudo labels on the Discrepancy Memorization, we pre-train the backbone segmentation network on the source images with the ground-truth segmentation masks (see Section 1 of the supplementary file). The pre-trained network yields more reliable pseudo labels, stabilizing the Discrepancy Memorization even in the initial stage.
>
> >### 5. There are no limitations to potential negative societal impact.
>
> Section 4 of the Supplementary Materials addresses the negative societal impacts. Our approach fosters comprehensive image analysis. However, the analysis may contain problematic information, possibly leading to the infringement of economic interests.

---

> ### Author Response · Authors · 2023-08-19
> **Sincerely Request Your New Comment**
>
> Dear Reviewer LZYW,
>
> We thank you again for your valuable comments, which significantly help us to polish our paper. We are looking to discussing with you the questions that are addressed unsatisfactorily.
>
> Best,
>
> Authors of Paper ID 10143

---

> > ### Author Response · Authors · 2023-08-21
> > **Sincerely Request Your New Comment Again**
> >
> > Dear Reviewer LZYW,
> >
> > Again, please allow us to extend our sincere thanks to you, for your time and efforts of reviewing our paper. As the deadline for the authors' response is approaching, we sincerely request your comment on our primary response. This will definitely give us a valuable chance to address the questions unsolved.
> >
> > Best,
> >
> > Authors of Paper ID 10143

---

### Author Rebuttal · Authors · 2023-08-08

## Response to Common Questions

>### **1. The advantage of style compensation based on the discrepancy features**
>- Reviewer LZYW-Q2 “Are there any advantages to storing these discrepancies rather than just performing style regression?”
>- Reviewer uGsB-Q2 “The proposed method is not specifically designed for open compound domain adaptation but rather for multi-target domain adaptation. How to generalize to unseen domains is not addressed.”
>- Reviewer okqW-Q8 “From a high-level, I doubt if this way can really help the segmentation on the open domain, as the compensation is designed for the target, i.e., based on the distribution or statistics of target images or features. How can such compensation generalize to an unseen domain?”

Below, we explain the advantage of our method of style compensation, which specifically transforms the object features of the target and open domains to the source domain. In Figure 1 of the rebuttal file, we conceptually compare the style compensation with the typical OCDA methods (e.g., ML-BPM [ECCV 2022], AST [AAAI 2022], CSFU [CVPR2021], DHA [NeurlPS2020], and CDAS [CVPR2020]) that optimize the parameters of the deep network to transform the object features of the source domain to the target domain, or vice versa.

*[Feature Transformation by Network Parameters]*

In Figure 1(a), we illustrate the source and target object features as the dots and crosses of different categories spanning over the latent space. The dot and cross of the same category represent distinct instances. The source and target features belong to separate clusters (the yellow as the source domain, the blue and red as the target domains) that indicate the source and target domains with distinct styles. The typical OCDA methods learn deep network parameters. The parameters (the dash arrows) transform the target object features into the source domain.

The parameters are learned from the source and target images in the training set. But they cannot trivially transform the open domain's object features  (see the dots and crosses in the green cluster) that contain many unseen styles. Using the learned parameters directly may transform the object features of the open domain to the latent sub-space misaligned with the source domain. A remedy is adaptively re-weighting and re-combining the learned parameters, which are generalized as the new parameters (denoted as the green arrow) to transform the open domain features. But the network contains many layers of parameters. How can the parameters correspond to the unknown number of target and open domains in different layers? How can we re-weight and re-combine these parameters without an explicit correspondence between the parameters and the target domains? These problems remain to be challenging. Moreover, we should not only transform the image-level features of the entire scenes but also the features of different object categories and instances, as we have discussed in the introduction of the main paper (see lines 30-36). This goal is hard to achieve without effectively generalizing the learned parameters.


*[Advantage of Style Compensation]*

Rather than using the network parameters to transform the target/open domain's features to the source domain, we learn the discrepancy features representing the style differences across domains. Figure 1(b) shows that we learn and store the discrepancy features. The discrepancy features (denoted as the solid blue and red arrows) explicitly correspond to different object categories and representative styles of instances. The stored discrepancy features are flexibly re-weighted and re-combined, based on the similarity between object features of target and open domains (see the weights $w_1$ and $w_2$ computed in Eq. (3)). In this way, we generalize the re-weighted discrepancy features. A new discrepancy feature (denoted as the green arrow) transforms the object features of the open domain to the source domain (see the formulation in Eq. (3) of the main paper). In Figure 1(c) of the rebuttal file, we visualize some of the object features of the open domain, which undergo style compensation or not. With the style compensation powered by the discrepancy features, we better transform the open domain’s object features into the source domain. In Table 7(c-f) of the main paper, we have evaluated our method on the target and open domains of different datasets, where our method outperforms other methods.


>### **2. Instance masks**
>- Reviewer LZYW-Q1 “It's unclear how different instances within the same class are differentiated.”
>- Reviewer uGsB-Q4 “The paper does not explain how to obtain instance masks.”
>- Reviewer okqW-Q3 “For the Instance-Key and Discrepancy Features, they are instance-level (see Eq. (2)). I just doubt if such method can be scalable to large-scale datasets, as the m may be super large (there are large numbers of pixels within one specific category).”
>- Reviewer CD2C-Q1 “You raise a valid point regarding the terminology Instance-Key which may not be appropriate in the context of semantic segmentation tasks where instance-level annotations are not available.”

We clarify that instance masks are unnecessary in our method. The term “instance” of “instance-key feature” does not strictly point to an object instance. An instance-key feature is updated by the same category of object features, which are extracted from different images. The object features from different images naturally represent various object instances. Thus, the instance-key feature is named as such to indicate it is sensitive to the object instances. More specifically, the instance-key feature summarizes the same category’s object instances with similar visual styles. Consequently, an instance-key feature captures one of the representative styles, which widely appear with the object instances of the same category. We plan to revise the “instance-key feature” to the “representative-key feature” to fix the inappropriate name.

---

### Author Response · Authors · 2023-08-17
**Sincerely Request Your New Comment**

Dear Reviewers,

We thank you again for your valuable comments, which significantly help us to polish our paper. Could we kindly know if the responses have addressed your concerns and if further explanations or clarifications are needed? Your time and efforts in evaluating our work are appreciated greatly.


Best,

Authors of Paper ID 10143

---

### Comment · Area_Chair_Nqi8 · 2023-08-21
**Final Rating Required**

Dear Reviewer okqW, LZYW, and XmuL,

Could you help share a quick feedback to authors' rebuttal and give your final rating? Thank you!

Best regards,
AC

---

> ### Author Response · Authors · 2023-08-21
> **Sincere Thanks to AC**
>
> Dear AC Nqi8,
>
> We deeply appreciate your support towards our work and your valuable contributions to the community.
>
> Best,
>
> Authors of Paper ID 10143

---

### Decision · Program_Chairs · 2023-09-21

**Decision:**

Accept (poster)

**Comment:**

Most reviewers generally found that the paper is well organized and easy to read; the proposed method is novel or interesting; and the proposed method is effective and achieves SOTA performance. However, there are two common questions: (1) How exactly are the object instances obtained, and what’s their influence on the scalability and limitation of the proposed method? (2) Why is storing style differences important, how can this method address the open compound domain adaptation problem, and how can it generalize to unseen domains? Authors provided detailed feedback on the two common questions. Specifically, for (1), they clarified that their method does not require object instance segmentation. Therefore, this question was due to unclear presentation of the paper and authors promised to modify the name of “instance” to “representation”. For (2), authors also provided a detailed explanation on the mechanism of the proposed method, which seems reasonable in addressing the generalization problem into adapting all features to source domain. There are also other concerns raised by reviewers, which have been carefully addressed by authors. After rebuttal, all ratings are positive (6, 6, 5, 5) except a negative score (3: Reject) by Reviewer okqW. Reviewer okqW provided a number of weakness points. Authors carefully addressed all these points. After rebuttal, Reviewer okqW provided feedback feeling still confused about the mechanism generalizing to unseen domains. AC assumes that except this concern, other concerns have been addressed. Authors further provided an explanation, which seems reasonable in addressing the concern. Considering all the above, the paper’s strengths overwhelm its weaknesses, and it shares some useful insights to the community. AC therefore is happy to accept the paper. Authors are required to address all reviewer comments and incorporate the rebuttal and discussion material to the camera-ready version of the paper.